# Deep Models, Shallow Alignment: Uncovering the Granularity Mismatch in Neural Decoding

## Abstract

Neural visual decoding is a central problem in brain–computer interface research, aiming to reconstruct human visual perception and to elucidate the structure of neural representations. However, existing approaches overlook a fundamental granularity mismatch between human and machine vision, where deep vision models emphasize semantic invariance by suppressing local texture information, whereas neural signals preserve an intricate mixture of low-level visual attributes and high-level semantic content. To address this mismatch, we propose Shallow Alignment, a contrastive learning strategy that aligns neural signals with intermediate representations of visual encoders rather than their final outputs, thereby striking a better balance between low-level texture details and high-level semantic features. Extensive experiments across multiple benchmarks demonstrate that Shallow Alignment significantly outperforms standard final-layer alignment, with performance gains ranging from 22% to 58% across diverse vision backbones. Notably, our approach reveals a positive scaling trend in neural visual decoding, enabling decoding performance to improve consistently with the capacity of pre-trained vision backbones. We further conduct systematic empirical analyses to shed light on the mechanisms underlying the observed performance gains. Code is available at `https://anonymous.4open.science/r/repo-7f41a8`.

## 1 Introduction

Understanding how the brain encodes visual information is a fundamental problem in both neuroscience and artificial intelligence (Hubel & Wiesel, 1968; Van Essen et al., 1992; Nauhaus et al., 2012; Zeiler & Fergus, 2014; Yamins & DiCarlo, 2016; Liang et al., 2018). Recent advances in brain-computer interfaces, particularly studies based on electroencephalography (EEG) and magnetoencephalography (MEG), have demonstrated the feasibility of decoding visual stimuli directly from neural activity (Spampinato et al., 2017; Grootswagers et al., 2022; Hebart et al., 2023; Song et al., 2025). This task, commonly referred to as neural visual decoding, aims to retrieve perceived visual stimuli from non-invasive neural recordings. The core challenge lies in learning an alignment function that can translate high-dimensional, noisy neural dynamics into structured visual representations.

The prevailing paradigm in neural visual decoding adopts large-scale pre-trained vision models (e.g., CLIP) as feature extractors, aligning neural signals to visual representations via contrastive learning on the final-layer embeddings of these models (Song et al., 2023; Li et al., 2024a; Zhang et al., 2025a; Wu et al., 2025; Zhang et al., 2025b). However, these approaches overlook a fundamental granularity mismatch between human and machine vision. In this work, granularity denotes the level of visual abstraction and spatial aggregation encoded by a visual representation, ranging from local perceptual details to globally aggregated semantic embeddings. The inductive bias of contemporary vision models aims to maximize semantic invariance, thereby suppressing local variations to optimize for categorization (Krizhevsky et al., 2012; Papyan et al., 2020; Khan et al., 2022). In contrast, visually evoked neural responses are not confined to a single level of abstraction, but encode information across multiple representational scales, ranging from low-level attributes such as contours, color, and spatial frequency to high-level semantic content (DiCarlo et al., 2012; Carlson et al., 2013; Cichy et al., 2014; Garg et al., 2019).

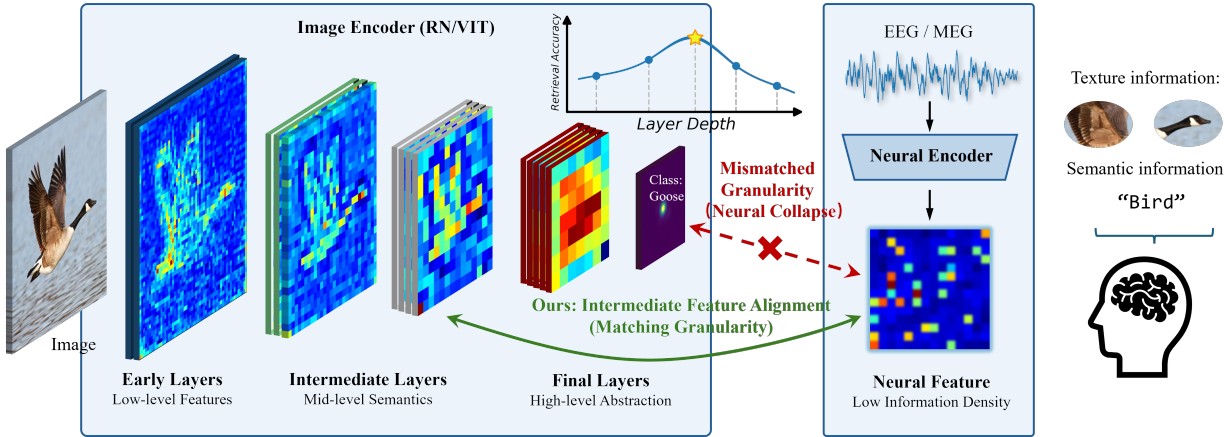

Figure 1: Overview of the proposed Shallow Alignment framework. The model aligns neural signals with intermediate visual representations to mitigate information-lossy alignment at high semantic levels.

Aligning neural signals to the final, most abstract layer of vision models forces a complex, multi-scale neural representation to match a texture-insensitive semantic embedding. An immediate empirical consequence of this mismatch is a counterintuitive scaling failure: under final-layer alignment, increasing the capacity of the vision backbone does not reliably improve decoding performance. In fact, simpler models such as ResNet-50 can outperform large-scale Vision Transformers (Wu et al., 2025; Zhang et al., 2025b), a phenomenon we term the *Depth–Capacity Paradox*. This paradox arises because larger models undergo more aggressive semantic compression in their final layers, thereby amplifying the granularity mismatch with neural signals. This inversion of the usual scale–performance relationship is not an incidental observation but a structural signal: under the current alignment protocol, additional model capacity becomes actively counterproductive.

Visually evoked EEG/MEG signals contain information at multiple levels of abstraction, including low-level perceptual attributes and higher-level semantic content. This suggests that an intermediate depth in the vision encoder, rather than its final output, may offer the closest structural correspondence to the multi-scale information present in neural signals. Motivated by this insight, we propose Shallow Alignment, a strategy that shifts the neural decoding target from the final output layer to the intermediate layers of vision encoders. We posit that intermediate representations offer a more appropriate granularity match for non-invasive neural signals, balancing high-level semantic abstraction with the retention of structural fidelity. We further examine granularity calibration from two complementary perspectives. Token-level calibration compares spatial pooling strategies to identify informative visual regions for neural alignment, while layer-level calibration uses multi-layer cascade fusion to combine complementary intermediate representations across depths. Shallow Alignment provides an empirical granularity-calibration protocol, rather than a fully automatic granularity-matching estimator.

Our contributions are summarized as follows:

- We propose Shallow Alignment, a neural visual decoding framework that mitigates the granularity mismatch between neural signals and vision encoders, yielding consistent improvements across multiple benchmarks.

- We resolve a Depth–Capacity Paradox, where increased semantic abstraction in deeper layers hinders effective alignment. By addressing this, we recover a positive scaling trend for neural decoding, enabling performance to improve with model capacity.

- A finer-grained analysis of granularity calibration is conducted through spatial pooling and multi-layer cascade fusion, showing that adaptive token weighting and multiple-layer aggregation further enhance alignment by matching the spatial and hierarchical granularity of neural responses.

## 2 Related Work

### 2.1 Semantic Granularity in Neural Decoding

Recent advancements in neural decoding have predominantly focused on aligning neural signals with the latent spaces of large-scale pre-trained vision models. Song et al. (2023) introduced a contrastive learning approach that incorporates plug-and-play self-attention and graph attention modules to capture spatial correlations in EEG electrodes for effective image decoding. However, emerging evidence suggests that maximizing semantic abstraction does not necessarily yield optimal decoding performance. For instance, Wu et al. (2025) identified a "System GAP" between human perception and digital stimuli, proposing an Uncertainty-Aware Blur Prior (UBP) that improves alignment by dynamically adjusting the blur radius based on sample uncertainty. Similarly, Zhang et al. (2025b) introduced NeuroBridge, which leverages Cognitive Prior Augmentation to simulate perceptual variability via image transformations, including Gaussian blur, Gaussian noise, mosaic effects, and low-resolution downsampling. Notably, they also reported a counter-intuitive phenomenon where architecturally simpler backbones (e.g., ResNet-50) often exhibit competitive or even superior performance compared to large-scale foundation models (e.g., ViT) under standard settings. This observation motivates our hypothesis that the key to alignment lies not in maximizing semantic level of the image or neural representation, but in precisely calibrating the representational granularity between modalities.

### 2.2 Hierarchical Representations in Human and Artificial Vision

Neuroscience has established that the primate visual cortex operates via a cascade of processing stages, where early areas (e.g., V1, V2) encode low-level features like spatial frequencies and orientation, while high-level object semantics emerge only in later stages (e.g., IT cortex) (DiCarlo et al., 2012; Cichy et al., 2014). An interesting parallelism exists in deep artificial neural networks, which evolve from detecting low-level primitives in shallow layers to representing abstract concepts in deep layers (Zeiler & Fergus, 2014; Mahendran & Vedaldi, 2015). In addition, non-invasive neural recordings (e.g., EEG/MEG) capture visual processing dynamics across multiple levels of granularity, and a substantial body of encoding and decoding work has mapped the CNN layer hierarchy onto this neural organization. Successive network layers have been shown to track successive cortical processing stages in fMRI during natural movie viewing (Wen et al., 2018) and in high-temporal-resolution MEG during object viewing (Seeliger et al., 2018). Simultaneous EEG–fMRI further reveals a similar hierarchical correspondence during rapid visual processing (Tu et al., 2018). Moreover, aligning multiple or non-final layers, rather than relying solely on final-layer outputs, has been shown to improve brain alignment across EEG, fMRI, and behavior (Lu et al., 2026). This supports the idea that intermediate representations can offer a more suitable balance between pictorial structure and semantic abstraction for neural decoding.

### 2.3 Granularity Balance in Intermediate Representations

Granularity can be evaluated on a trained network, where prior work provides complementary, layer-wise techniques for characterizing both semantic and spatial granularity. Previous studies have shown that visual representations progress from low-level colors, edges, and textures to higher-level parts and objects (Zeiler & Fergus, 2014; Bau et al., 2017). Using centered kernel alignment, attention distance, and effective receptive fields, Raghu et al. (2021) revealed the spatial aggregation of visual features across layers, which grows gradually in CNNs but already incorporates substantial global information in the lower layers of Vision Transformers. Prior research has also demonstrated the utility of intermediate layers for diverse tasks, ranging from elucidating training dynamics (Alain & Bengio, 2016) to enhancing transfer learning (Evci et al., 2022) and improving robustness against distribution shifts (Lee et al., 2022; Uselis & Oh, 2025). Crucially, these works point to a fundamental trade-off in feature evolution. As networks deepen, they undergo Neural Collapse (Papyan et al., 2020), where class representations inevitably collapse toward their mean centers in the final layer to maximize separability, filtering out the structural details. We argue that this characteristic makes intermediate layers, rather than the collapsed final output, superior alignment target for neural visual decoding.

## 3 Methodology

As illustrated in Figure 1, we propose Shallow Alignment, which mitigates the granularity mismatch between neural signals and visual representations.

### 3.1 Problem Formulation

We formulate neural visual decoding as a cross-modal representation alignment problem between neural signals and visual stimuli. Let $\mathcal{D} = (x_N^{(k)}, x_I^{(k)})_{k=1}^{M}$ denote a paired dataset of $M$ samples, where $x_N^{(k)} \in \mathbb{R}^{C \times T}$ represents a non-invasive neural signal with $C$ channels and $T$ time points, and $x_I^{(k)} \in \mathbb{R}^{H \times W \times 3}$ denotes the corresponding visual stimulus. The objective is to learn a neural encoder $f_\theta : \mathcal{X}_N \to \mathcal{Z}$ that maps raw neural signals into a latent embedding space $\mathcal{Z}$, aligning them with visual representations extracted by a pre-trained vision encoder $E_\phi$. We employed $f_\theta$ as EEGProject (Wu et al., 2025), a parameter-efficient MLP in which the flattened signal $x_N \in \mathbb{R}^{C \cdot T}$ is linearly projected to a $d$-dimensional latent space and refined by a residual nonlinear block. In conventional approaches, the neural embedding $z_N = f_\theta(x_N)$ is aligned with the final-layer output of the vision encoder, denoted as $z_{\text{last}} = E_\phi(x_I)$. However, the final-layer representation $z_{\text{last}}$ is typically a highly compressed and semantically abstract embedding in which local structural information is largely suppressed. Aligning such representations with noisy and multi-scale neural signals exacerbates a granularity mismatch between these two modalities.

### 3.2 Pre-trained Vision Models

For the vision encoder $E_\phi$, we consider a diverse set of pre-trained visual backbones, ranging from conventional convolutional architectures to large-scale Vision Transformers. Specifically, we include ResNet-50 and ResNet-101 (He et al., 2016), as well as Vision Transformer models of increasing capacity, including ViT-B/16 (Dosovitskiy et al., 2021), ViT-H/14 (Zhai et al., 2022), and ViT-bigG/14 (Cherti et al., 2023), all using pretrained weights provided by OpenCLIP (Ilharco et al., 2021). To systematically study the effect of model scale and architecture on neural visual decoding, we further incorporate several state-of-the-art large-scale visual encoders, including DINOv2 (Oquab et al., 2024), EVA-02 (Fang et al., 2023), and InternViT (Chen et al., 2024). For relatively shallow architectures (e.g., ResNet), we evaluate representations from all layers. For ultra-deep Transformer-based models, we adopt a uniform sampling strategy, probing approximately ten evenly spaced intermediate layers to estimate layer-wise performance. Based on the sweep results, we then select the layer whose representations are best suited to EEG/MEG signals.

### 3.3 Shallow Alignment via Intermediate Representations

To mitigate the granularity mismatch, we introduce Shallow Alignment, which shifts the alignment target from the final output to intermediate representations of the vision encoder. Consider a deep vision encoder $E_\phi$ composed of $L$ layers. Given an input image $x_I$, the encoder produces a sequence of hidden representations $\mathbf{H} = \{h^{(1)}, h^{(2)}, \ldots, h^{(L)}\}$, where $h^{(l)}$ denotes the feature map at the $l$-th layer.

In contrast to the final embedding $z_{\text{last}}$, which is optimized for semantic invariance and is largely insensitive to local spatial variations, intermediate representations capture a more balanced level of abstraction. In particular, the representation at a selected depth $l^*$ preserves informative structural details while remaining sufficiently discriminative at the semantic level.

Formally, we define the target visual embedding $z_I$ as the pooled feature from the intermediate layer $l^*$:

$$z_I = \text{Pool}\Big(h^{(l^*)}(x_I)\Big), \quad 1 \leq l^* < L \tag{1}$$

where $\text{Pool}(\cdot)$ denotes a spatial aggregation operation that maps patch-level features into a fixed-dimensional vector, including global average, center-region, and adaptive attention-based pooling for token-level granularity calibration.

### 3.4 Linear Semantic Projector

To align the two modalities, we project the neural features $z_N$ and visual features $z_I$ into a shared latent space using linear mappings:

$$\mathbf{v} = W_N z_N + b_N, \quad \mathbf{w} = W_I z_I + b_I, \tag{2}$$

where $W_N$ and $W_I$ are learnable projection matrices, and $b_N$ and $b_I$ denote bias terms.

These linear transformations act as learnable projections that distill high-dimensional, redundant intermediate features into a latent subspace aligned with the neural manifold. By restricting the projection to be linear, we explicitly limit model capacity, encouraging the alignment performance to stem from the quality of the intermediate representations themselves rather than from expressive but potentially overfitting decoders.

### 3.5 Contrastive Objective

We employ a symmetric contrastive objective (Radford et al., 2021) that encourages matched neural–visual pairs $(\mathbf{v}^{(k)}, \mathbf{w}^{(k)})$ to exhibit high similarity, while separating mismatched pairs within each mini-batch. The contrastive loss is defined as

$$\mathcal{L}_C = -\frac{1}{2M} \sum_{k=1}^{M} \left[ \log \frac{\exp\big(\mathrm{sim}(\mathbf{w}^{(k)}, \mathbf{v}^{(k)})/\tau\big)}{\sum_{j=1}^{M} \exp\big(\mathrm{sim}(\mathbf{w}^{(k)}, \mathbf{v}^{(j)})/\tau\big)} \right. $$
$$ \left. + \log \frac{\exp\big(\mathrm{sim}(\mathbf{v}^{(k)}, \mathbf{w}^{(k)})/\tau\big)}{\sum_{j=1}^{M} \exp\big(\mathrm{sim}(\mathbf{v}^{(k)}, \mathbf{w}^{(j)})/\tau\big)} \right], \tag{3}$$

where $\mathrm{sim}(\cdot, \cdot)$ denotes cosine similarity, $\tau$ is a temperature hyperparameter, and $M$ denotes the number of paired samples in the batch. This bidirectional formulation enforces consistent alignment across modalities and is used as the primary training objective (Wang et al., 2019).

## 4 Experiments

### 4.1 Datasets

**THINGS-EEG** (Gifford et al., 2022) provides 63-channel electroencephalography (EEG) recordings from 10 participants collected under a Rapid Serial Visual Presentation (RSVP) paradigm (Grootswagers et al., 2019). It comprises a training set of 1,654 unique object concepts and a disjoint test set of 200 concepts. For training, each concept is associated with 10 distinct images, each presented 4 times. In the test set, a single image is used per concept and repeated 80 times.

**THINGS-MEG** (Hebart et al., 2023) contains 271-sensor magnetoencephalography (MEG) recordings from 4 participants (500 ms stimulus presentation, $1000 \pm 200$ ms ITI) covering 1,854 object concepts, partitioned into 1,654 training and 200 testing concepts. Each training concept includes 12 distinct images, while each test concept consists of 12 repetitions of a single image.

Signals are preprocessed following prior work (Wu et al., 2025; Zhang et al., 2025b): bandpass filtered (0.1–100 Hz for EEG, 0.1–40 Hz for MEG), epoched 0–1000 ms post-stimulus with 200 ms pre-stimulus baseline correction, downsampled (250 Hz for EEG, 200 Hz for MEG), and averaged across repetitions. We restrict analysis to occipital and parietal channels overlying visual cortex. Full details in Appendix A.1.

### 4.2 Implementation Details

All experiments are implemented in PyTorch and conducted on NVIDIA RTX A6000 GPUs. We utilize EEGProject as the neural encoder and select the channels corresponding to the overlying occipital and parietal cortex related to human visual processing. Models are trained for 50 epochs with a batch size of 1,024 using the AdamW optimizer, with a learning rate of $1 \times 10^{-4}$ and a weight decay of $1 \times 10^{-4}$. For evaluation, we compute retrieval accuracy using cosine similarity between neural and image embeddings, reporting Top-1 and Top-5 accuracies under intra-subject and inter-subject settings. For fair comparison

Table 1: Overall accuracy (%) of 200-way zero-shot retrieval on THINGS-EEG: Top-1 and Top-5.

| Method | Metric | Sub1 | Sub2 | Sub3 | Sub4 | Sub5 | Sub6 | Sub7 | Sub8 | Sub9 | Sub10 | Avg. |
|---|---|---|---|---|---|---|---|---|---|---|---|---|
| **Intra-Subject: train and test on one subject** | | | | | | | | | | | | |
| NICE | Top-1 | 13.2 | 13.5 | 14.5 | 20.6 | 10.1 | 16.5 | 17.0 | 22.9 | 15.4 | 17.4 | 16.1 |
| | Top-5 | 39.5 | 40.3 | 42.7 | 52.7 | 31.5 | 44.0 | 42.1 | 56.1 | 41.6 | 45.8 | 43.6 |
| ATM | Top-1 | 25.6 | 22.0 | 25.0 | 31.4 | 12.9 | 21.3 | 30.5 | 38.8 | 34.4 | 29.1 | 27.1 |
| | Top-5 | 60.4 | 54.5 | 62.4 | 60.9 | 43.0 | 51.1 | 61.5 | 72.0 | 51.5 | 63.5 | 58.1 |
| Neural-MCRL | Top-1 | 27.5 | 28.5 | 37.0 | 35.0 | 22.5 | 31.5 | 31.5 | 42.0 | 30.5 | 37.5 | 32.4 |
| | Top-5 | 64.0 | 61.5 | 69.0 | 66.0 | 51.5 | 61.0 | 62.5 | 74.5 | 59.5 | 71.0 | 64.1 |
| UBP | Top-1 | 41.2 | 51.2 | 51.2 | 51.1 | 42.2 | 57.5 | 49.0 | 58.6 | 45.1 | 61.5 | 50.9 |
| | Top-5 | 70.5 | 80.9 | 82.0 | 76.9 | 72.8 | 83.5 | 79.9 | 85.8 | 76.2 | 88.2 | 79.7 |
| NeuroBridge | Top-1 | 50.0 | 63.2 | 61.6 | 61.4 | 54.8 | 69.7 | 62.7 | 71.2 | 64.0 | 73.6 | 63.2 |
| | Top-5 | 77.6 | 90.6 | 91.1 | 90.0 | 85.0 | 92.9 | 88.8 | 95.1 | 91.0 | 97.1 | 89.9 |
| Ours | Top-1 | 75.0 | 87.5 | 83.2 | 79.5 | 74.6 | 89.9 | 78.5 | 86.9 | 81.3 | 89.3 | 82.6 |
| | Top-5 | 94.3 | 98.9 | 98.2 | 96.1 | 96.4 | 99.3 | 97.3 | 99.4 | 97.8 | 99.1 | 97.7 |
| **Inter-Subject: leave one subject out for test** | | | | | | | | | | | | |
| NICE | Top-1 | 7.6 | 5.9 | 6.0 | 6.3 | 4.4 | 5.6 | 5.6 | 6.3 | 5.7 | 8.4 | 6.2 |
| | Top-5 | 22.8 | 20.5 | 22.3 | 20.7 | 18.3 | 22.2 | 19.7 | 22.0 | 17.6 | 28.3 | 21.4 |
| ATM | Top-1 | 10.5 | 7.1 | 11.9 | 14.7 | 7.0 | 11.1 | 16.1 | 15.0 | 4.9 | 20.5 | 11.9 |
| | Top-5 | 26.8 | 24.8 | 33.8 | 39.4 | 23.9 | 35.8 | 43.5 | 40.3 | 22.7 | 46.5 | 33.8 |
| Neural-MCRL | Top-1 | 13.0 | 12.0 | 14.5 | 12.5 | 11.5 | 13.5 | 14.0 | 18.5 | 13.5 | 17.0 | 14.0 |
| | Top-5 | 31.5 | 30.5 | 35.5 | 35.5 | 29.0 | 35.5 | 36.0 | 38.5 | 32.5 | 39.0 | 34.3 |
| UBP | Top-1 | 11.5 | 15.5 | 9.8 | 13.0 | 8.8 | 11.7 | 10.2 | 12.2 | 15.5 | 16.0 | 12.4 |
| | Top-5 | 29.7 | 40.0 | 27.0 | 32.3 | 33.8 | 31.0 | 23.8 | 32.2 | 40.5 | 43.5 | 33.4 |
| NeuroBridge | Top-1 | 23.2 | 21.2 | 13.2 | 17.0 | 14.5 | 25.0 | 15.3 | 20.1 | 13.7 | 27.2 | 19.0 |
| | Top-5 | 52.4 | 49.3 | 36.5 | 45.3 | 37.7 | 55.0 | 45.1 | 44.9 | 36.5 | 56.3 | 45.9 |
| Ours | Top-1 | 23.5 | 30.6 | 10.0 | 19.5 | 18.1 | 22.7 | 18.6 | 17.3 | 23.0 | 34.4 | 21.8 |
| | Top-5 | 53.2 | 60.4 | 28.1 | 48.3 | 45.2 | 49.8 | 46.0 | 46.1 | 54.8 | 62.0 | 49.4 |

with prior baselines, we follow the dataset protocol by training on the provided training split and evaluating on the held-out zero-shot test split. Appendix B.2 further verifies that validation-based checkpoint selection yields only minor performance differences while preserving the same intermediate-layer advantage. Reported results are averaged over five independent runs with different random seeds.

### 4.3 Bridging Cross-Modal Granularity Mismatch via Intermediate-Layer Alignment

We compared our Shallow Alignment strategy against state-of-the-art methods, including NICE (Song et al., 2023), ATM (Li et al., 2024a), Neural-MCRL (Li et al., 2024b), UBP (Wu et al., 2025), NeuroBridge (Zhang et al., 2025b), on both THINGS-EEG and THINGS-MEG datasets.

As shown in Table 1 and Table 2, our method achieves substantial improvements across all evaluation metrics. These results correspond to the best-performing layer of the best-performing model (InternViT). We report this peak to illustrate the potential of the method and underscore that the choice of alignment target has a decisive impact on retrieval performance. The full layer-wise behavior across all evaluated models is presented in Fig. 2. We organize existing approaches into three categories based on how they handle the granularity relationship between neural signals and visual representations, providing a unified perspective on the evolution of neural decoding methods.

**Semantic-focused alignment.** Baseline approaches such as NICE, ATM, and Neural-MCRL primarily aim to enhance alignment with high-level semantic representations, while largely neglecting the granularity mismatch between neural signals and visual features, potentially limiting retrieval performance. These methods achieve average Top-1 accuracies of 16.1%, 27.1%, and 32.4%, respectively.

**Implicit granularity adaptation.** UBP and NeuroBridge improve performance to 50.9% and 63.2% Top-1 accuracy, respectively, by incorporating data augmentation strategies. Although these gains are commonly attributed to improved robustness against perceptual variability and low-level acquisition noise, the observed improvements can also be interpreted as arising from implicit granularity adaptation. By blurring or perturbing images, these methods attenuate fine-grained texture details. This reduction in visual complexity shifts the feature manifold to a coarse granularity, thereby enabling a more robust match with the coarse and noisy neural recordings. However, this improved alignment comes at the expense of high-fidelity information for accurate neural decoding.

**Explicit granularity calibration.** In contrast, our method explicitly aligns neural signals with intermediate visual representations, exploiting the fact that these representations retain both structural and semantic visual information. Rather than degrading visual inputs to match neural signal granularity, Shallow Alignment selects the representation depth at which the vision encoder naturally produces features whose granularity matches that of neural signals. This design preserves informative structural details while achieving appropriate cross-modal alignment, resulting in a Top-1 accuracy of 82.6% under the intra-subject retrieval setting. The progression from semantic-focused methods (16.1–32.4%) through implicit adaptation (50.9–63.2%) to explicit calibration (82.6%) underscores that granularity alignment is a central factor governing neural decoding performance. Further experiments on the choice of EEG encoder and vision backbone are reported in Appendix B.3.

Under the inter-subject setting, our method yields an average Top-1 accuracy of 21.8%, representing a 2.8% improvement over NeuroBridge (19.0%). Although this confirms that intermediate-layer alignment remains beneficial beyond the intra-subject regime, the improvement is smaller than that observed in the intra-subject setting (19.4%). This is expected because inter-subject decoding introduces additional variability across individuals. Beyond neural distribution shift, different subjects may also exhibit distinct representational granularity in their neural responses, making it more difficult for a single visual alignment target to match all held-out subjects equally well. Nevertheless, the appendix B.1 comparison that only changes the visual alignment target shows robust improvements from final-layer to intermediate-layer alignment, indicating that the proposed granularity-matched alignment can still generalize to the inter-subject setting.

A consistent trend is observed on the THINGS-MEG dataset (Table 2). Under the intra-subject protocol, our method achieves 48.0% Top-1 accuracy and 74.4% Top-5 accuracy, substantially outperforming NeuroBridge (32.2% / 60.8%) and UBP (26.7% / 55.2%). Nevertheless, performance remains low for all methods in the inter-subject setting, which may reflect a more pronounced between-subject granularity mismatch in MEG recordings.

## 4.4 Recovering a Scaling Trend via Granularity-Matched Alignment

We extend our evaluation across a diverse set of vision backbones that vary in architecture, training objectives, and model scale. The results are summarized in Figure 2. We use the total parameter count of the pretrained vision backbone as the measure of model scale, because intermediate representations are learned as part of the full end-to-end pretrained model. We also verify the robustness of this analysis using effective parameters up to the selected intermediate layer.

Across models, the Top-1 retrieval accuracy exhibits a consistent inverted U-shaped trend as a function of layer depth, first increasing and then declining. This behavior aligns with the hierarchical nature of visual representations in deep networks, where early layers capture low-level texture information and deeper layers progressively encode more abstract semantic concepts. As a result, representations at different depths correspond to different degrees of semantic–texture entanglement. We argue that peak performance occurs when the granularity of the visual feature mixture most closely matches the inherent characteristics of neural signals, creating an optimal bridge for alignment. As illustrated in Figure 2(a), different models possess unique feature extraction capabilities, leading to distinct dynamic balances between texture and semantic information across their layers. For instance, ViT-H/14 achieves peak performance at approximately 35.5% of its relative depth, whereas InternViT peaks at around 60%. Detailed results are provided in Appendix B.1, with more comprehensive evaluations of EEG channel selection (Appendix B.4) and subject-level consistency (Appendix B.5).

Table 2: Overall accuracy (%) of 200-way zero-shot retrieval on THINGS-MEG: Top-1 and Top-5.

| Method | Metric | Sub1 | Sub2 | Sub3 | Sub4 | Avg. |
|---|---|---|---|---|---|---|
| **Intra-subject: train and test on the same subject** | | | | | | |
| UBP | Top-1 | 15.0 | 46.0 | 27.3 | 18.5 | 26.7 |
| | Top-5 | 38.0 | 80.5 | 59.0 | 43.5 | 55.2 |
| NeuroBridge | Top-1 | 16.5 | 53.7 | 40.4 | 18.1 | 32.2 |
| | Top-5 | 41.6 | 85.3 | 73.2 | 43.1 | 60.8 |
| Ours | Top-1 | 25.5 | 81.9 | 56.0 | 28.6 | 48.0 |
| | Top-5 | 54.5 | 97.4 | 87.5 | 58.3 | 74.4 |
| **Inter-subject: leave-one-subject-out (LOSO)** | | | | | | |
| UBP | Top-1 | 2.0 | 1.5 | 2.7 | 2.5 | 2.2 |
| | Top-5 | 5.7 | 17.2 | 10.5 | 8.0 | 10.4 |
| NeuroBridge | Top-1 | 4.3 | 3.6 | 3.0 | 2.5 | 3.4 |
| | Top-5 | 13.1 | 15.6 | 11.2 | 11.3 | 12.8 |
| Ours | Top-1 | 1.3 | 6.6 | 5.4 | 1.5 | 3.7 |
| | Top-5 | 6.9 | 18.5 | 18.5 | 7.9 | 13.0 |

As shown in Figure 2(b), conventional alignment using only the final output layer reveals a counterintuitive trend: increasing model size does not necessarily improve neural decoding performance. In fact, large-scale models often underperform due to excessive semantic abstraction at their final layers. For example, DINOv2 achieves a Top-1 accuracy of only 17.5% when aligned at its final layer, substantially lower than the performance of the much smaller ResNet-50 (40.3%). This observation indicates that highly compressed and invariant final-layer representations are poorly matched to neural signals, as they discard fine-grained structural information. In contrast, selecting an appropriate intermediate layer fundamentally alters this behavior. When alignment is performed at the optimal depth, a clear scaling trend emerges: decoding performance consistently improves with increasing model capacity. This scaling trend is robust to the definition of model scale: both the total backbone parameters, reflecting the overall capacity of the pretrained model, and the effective parameters up to the selected intermediate layer, reflecting the capacity actually used at inference, yield the same significant positive correlation (see Figure 2(c) and Appendix B.6). The most significant gain is observed in DINOv2, which improves by 58.4% when shifting the alignment target from the final output to its optimal intermediate representation.

The emergence of a consistent positive scaling trend under intermediate-layer alignment carries significant implications for the field of neural visual decoding. Under the conventional final-layer alignment paradigm, the absence of a scaling relationship effectively removes the incentive to adopt larger and more powerful vision models. Practitioners have no principled basis for expecting that a larger model will yield better decoding performance. Our findings demonstrate that the primary bottleneck in neural decoding is not the lack of visual feature quality in large models, but the incorrect selection of the alignment target. By calibrating the granularity mismatch, our method can effectively leverage the representational capacity of large-scale models.

### 4.5 Revealing Performance Trade-offs via Concept Analysis

We report *Concept Accuracy*, defined as the proportion of Top-5 retrieved images that share the same concept category (animals, food, vehicles, tools, clothing, or others) as the query, excluding the query image itself. Formally, for each query, we count the number of concept-matched items within the Top-5 results and normalize by $5M$, where $M$ denotes the number of queries.

As illustrated in Figure 3(b), layer-wise analysis reveals a clear divergence between concept accuracy and retrieval accuracy. Concept accuracy increases monotonically with network depth, reflecting progressively stronger semantic abstraction. In contrast, Top-1 retrieval accuracy follows a non-monotonic trend, peaking

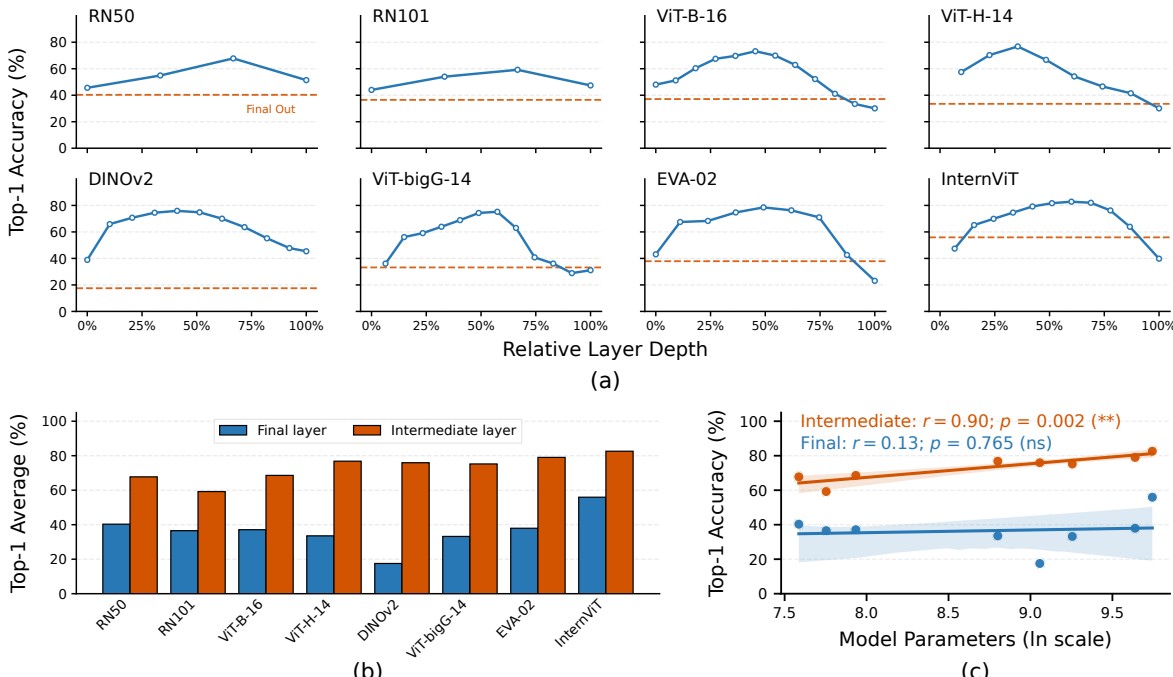

Figure 2: Comparative analysis of representational performance between intermediate and final layers on THINGS-EEG. (a) Top-1 accuracies of intermediate features across vision backbones. Relative depth is computed as $(\ell-1)/(L-1)$. Dashed orange lines mark the Final Output performance. For ResNet models, the final feature is obtained by attention pooling of the last convolutional layer. For Transformer-based models, the final feature corresponds to the CLS token embedding from the last layer. (b) Performance comparison across architectures. The bar chart summarizes the Top-1 accuracy gap between the best-performing intermediate layer (orange) and the final output layer (blue). (c) Scaling analysis. Linear regression analysis reveals the relationship between model scale (number of parameters in ln scale) and Top-1 accuracy. Statistical significance is denoted by asterisks (** for $p < 0.01$) or by "ns" for non-significant results ($p > 0.05$).

at an intermediate layer and declining sharply at the final layer. This behavior aligns with observations from the human visual system, which integrates mid-level visual cues (e.g., contours and texture) together with high-level semantic information, rather than collapsing representations purely to category identity. As a result, improvements in semantic consistency alone do not guarantee better retrieval performance. Instead, retrieval accuracy is maximized when visual representations preserve fine-grained structural details while maintaining sufficient semantic coherence.

Figure 3(c) presents representative examples of the Top-5 retrieval results. The ostrich query provides a revealing contrast between the two alignment strategies. Under intermediate-layer alignment, the Top-1 result correctly retrieves an ostrich image, while the remaining results, such as a baby crib and a table, are semantically unrelated to the query yet share a salient structural feature, slender supporting legs. This indicates that intermediate representations jointly encode high-level semantic information that correctly identifies the ostrich and mid-to-low-level structural patterns capturing the slender vertical shape, such that retrieval reflects a composite matching across multiple levels of visual abstraction. In contrast, final-layer alignment yields results dominated almost entirely by animals (e.g., sheep, pigeons, dogs), indicating that the representation has been compressed to category-level semantics in which structural cues such as shape and contour have been largely discarded. Although these results exhibit greater semantic category consistency, the over-reliance on category-level identity limits the ability to distinguish between individual instances. More retrieval results are provided in Appendix B.10.

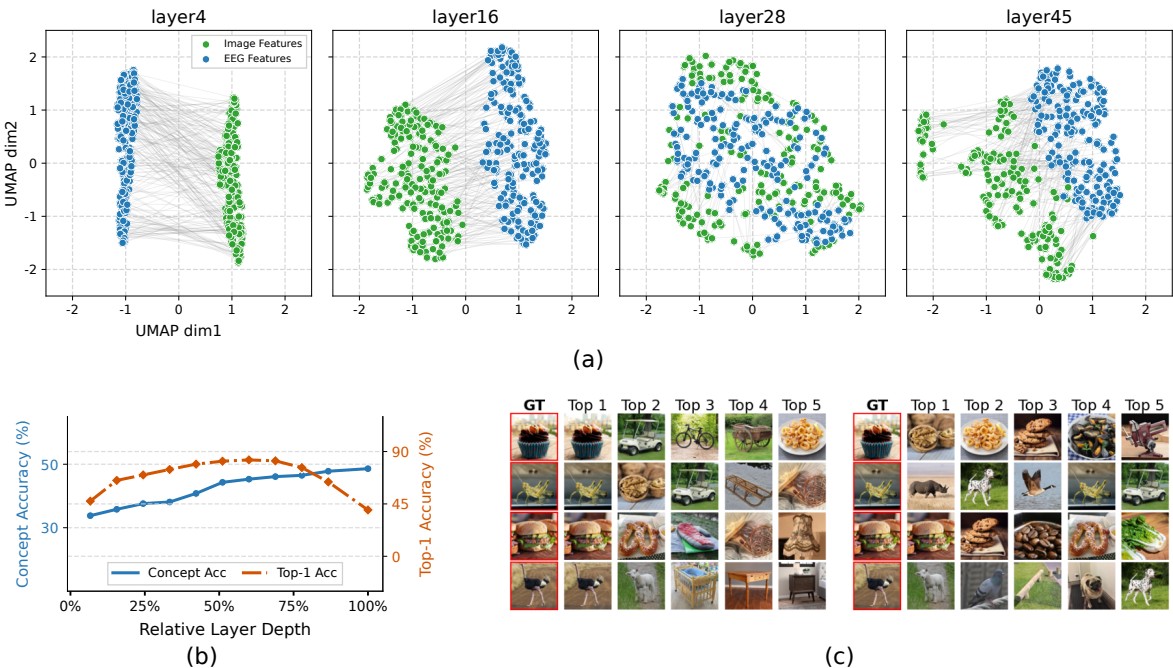

(a)

(b)

(c)

Figure 3: Analysis of layer-wise cross-modal alignment on THINGS-EEG (Subject 7). (a) UMAP visualization of feature alignment between EEG embeddings and visual representations extracted from selected intermediate layers of InternViT on the test set ($M = 200$). Gray lines connect ground-truth image–EEG pairs. (b) Concept accuracy and retrieval Top-1 accuracy versus relative layer depth, computed as $(\ell - 1)/(L - 1)$. (c) Top-5 retrieved samples using the best intermediate-layer embeddings (left) and final output embeddings (right). The red box indicates the ground-truth target.

These results indicate that the performance degradation observed at the final layer is not caused by insufficient semantic representation, but by the loss of texture-level information in highly abstract visual features. By retaining such details, intermediate representations achieve a more favorable balance between semantic correctness and discriminative precision.

## 4.6 Visualizing Granularity Consistency via Manifold Distributions

We employ UMAP (McInnes et al., 2018) to visualize the geometric distributions of the projected neural embeddings **v** and visual embeddings **w** on the test set, by stacking them as samples in a shared embedding space and fitting a single UMAP on the combined set, so that each point in the 2D plot corresponds to an individual **v** or **w**.

As shown in Figure 3(a) and the extended results in Appendix B.8, the layer-wise UMAP visualizations reveal a characteristic pattern as a function of network depth. Across layers, the degree of cross-modal mixing follows a clear trajectory that mirrors the above retrieval performance curve. At early layers, the EEG and visual embeddings form clearly separated clusters with distinct boundaries, and the gray lines connecting paired samples span large distances across the two clusters. At intermediate layers, the two modalities substantially overlap and intermingle in the projected space, with paired samples positioned in close proximity. As the network deepens toward the final output layer, the modality separation re-emerges and paired distances increase again. This progression from separation to mixing and back to separation constitutes a U-shaped modality gap that is the geometric counterpart of the inverted U-shaped retrieval accuracy curve observed in Figure 2(a). The layer at which cross-modal mixing is maximized coincides with the layer at which retrieval performance peaks, providing independent geometric evidence that optimal alignment occurs at the granularity-matched intermediate depth. When alignment is performed at the early layer or the final output layer, the embeddings from the two modalities form clearly separated clusters with

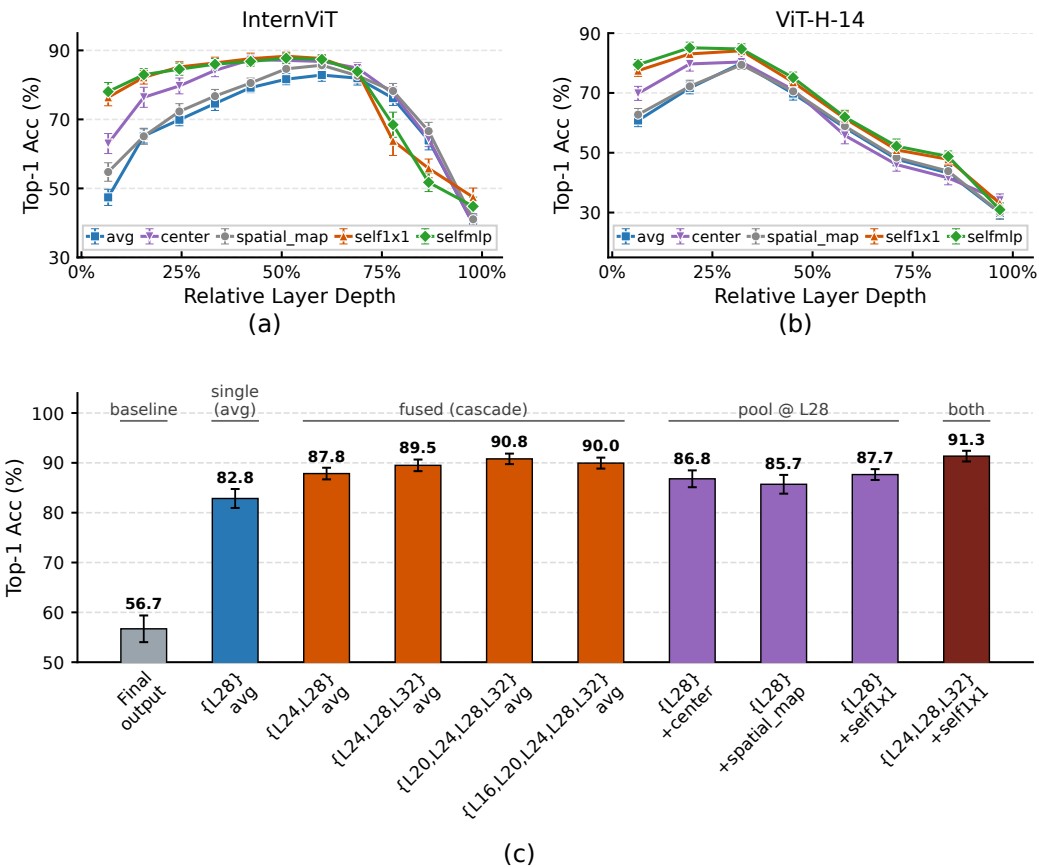

Figure 4: Ablation analysis of layer selection, spatial pooling, and cascade fusion on THINGS-EEG. (a) Retrieval Top-1 accuracy of InternViT versus relative layer depth under five spatial pooling strategies: *avg* (global mean over all patches), *center* (mean over the central $16 \times 16$ foveal region of the $32 \times 32$ patch grid), *spatial_map* (a learned softmax map shared across images), *self1x1* (per-image attention from a $1{\times}1$ linear scorer), and *selfmlp* (per-image attention from a two-layer MLP scorer). Results are averaged over 10 subjects; error bars denote SEM. (b) Same comparison for ViT-H-14. (c) Component ablation on InternViT features (THINGS-EEG intra-subject). We compare the final ViT output, the best single layer $\{L_{28}\}$, multi-layer retrieval-matrix fusion over neighboring intermediate layers, different pooling strategies at $L_{28}$, and fusion with self-$1 \times 1$ pooling.

distinct boundaries. This separation reflects a pronounced modality gap, indicating that the granularity of visual representations at both early and final layers is poorly matched to that of neural signals. In contrast, at the proper intermediate layer, EEG and visual embeddings substantially overlap and intermingle in the projected space. This observation suggests that intermediate representations induce a manifold structure whose granularity is more consistent with neural signals, thereby supporting more effective alignment.

## 4.7 Assessing Pooling Strategies for Cross-Modal Alignment

We investigate spatial pooling strategies more systematically. Although global average pooling is computationally efficient and parameter-free, it implicitly assumes that every patch contributes equally to the aligned embedding. However, f oveal regions at the image centre, where the photographic subject typically resides, are likely over-represented in the neural signal yet occupy only a small fraction of the patch grid relative to background patches. Uniform averaging therefore dilutes the foreground signal with task-irrelevant context.

To quantify this effect, we compare five pooling heads spanning three design categories. The parameter-free baselines include global mean pooling over all patches and mean pooling restricted to the central patches. A shared spatial weighting scheme learns a single softmax map applied across all images. Sample-adaptive heads compute per-image attention weights from a 1×1 linear projection and a two-layer MLP, respectively.

As shown in Figure 4a–b, adaptive weighting consistently outperforms uniform pooling, with the gain most pronounced in shallow layers. At an early layer of InternViT, accuracy rises from 47.4% (avg) to 78.1% (selfmlp). This behavior reflects the locality of early ViT tokens, which have undergone only a few rounds of self-attention and have not yet integrated global object identity. Uniform averaging in this regime dilutes the few informative patches with many uninformative ones, while a learned re-weighting can recover the lost signal. Notably, the shared map brings only marginal improvement over average map (for example, only an increase of 7.4% at the early layer of InternViT), because the discriminative patch distribution varies substantially across stimuli, and a single global map cannot capture the salient regions of each image. Sample-adaptive heads that produce input-dependent weights are required to realise the full gain.

A complementary view emerges from comparing center-restricted pooling with global average pooling, which confirms the centrality hypothesis. Simply restricting the pool to the central quarter of patches lifts accuracy at early layers of InternViT and ViT-H-14 by 15.6% and 9.0%, respectively, with the gap monotonically shrinking with depth and vanishing at the deepest layers. This is consistent with global attention progressively propagating object information to all patches, so that spatial selection becomes irrelevant once features are fully global.

### 4.8 Aggregating Complementary Granularity via Multi-Layer Cascade Fusion

Since different intermediate layers encode complementary granularity, we examine whether aggregating several layers yields additional gains. For each candidate layer, we independently train a neural encoder, compute its cross-modal similarity matrix, and fuse the predictions by averaging these matrices. The fused layers are selected as the top-performing layers in the single-layer evaluation, clustered around the optimal depth $L_{28}$.

As shown in Figure 4(c), retrieval-matrix fusion consistently improves over single-layer alignment. Using only $\{L28\}$ with average pooling achieves 82.8% Top-1 accuracy, while fusing $\{L24, L28\}$ increases performance to 87.8%. Incorporating additional neighboring layers further improves accuracy to 89.5% and 90.8%, suggesting that adjacent intermediate representations encode partially complementary structural and semantic information. However, the improvement becomes progressively smaller as more layers are included, and expanding the fusion to five layers slightly reduces performance to 90.0%. This indicates a clear marginal effect: multi-layer fusion benefits from broader granularity coverage, but overly broad aggregation may introduce redundant or less well-matched representations. The best result is achieved when retrieval-matrix fusion is combined with adaptive self-1 × 1 pooling (91.3%), indicating that spatial re-weighting further enhances the fused multi-level representation.

## 5 Conclusion

In this work, we identifies a fundamental granularity mismatch in neural visual decoding: non-invasive neural signals preserve multi-scale visual information, whereas the final layers of deep vision models collapse representations into highly abstract semantic embeddings. Shallow Alignment mitigates this mismatch by aligning EEG/MEG signals with intermediate visual representations, which better balance structural details and semantic abstraction. This granularity-matched alignment substantially improves retrieval performance and recovers a positive scaling trend with model capacity.

Our analyses further show that granularity calibration extends beyond selecting a single intermediate layer. Spatial pooling adjusts token-level granularity by emphasizing informative visual regions, while multi-layer cascade fusion aggregates neighboring layers with complementary structural–semantic information. We note that Shallow Alignment is a granularity-calibration protocol based on layer selection, not an automatic granularity-matching method, leaving a principled automatic estimator to future work.

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

# A    Experimental Details

## A.1    Datasets for Experiments

**THINGS-EEG.**    We conduct experiments on the THINGS-EEG dataset, which contains electroen-cephalography (EEG) recordings from 10 human subjects performing a visual recognition task under a Rapid Serial Visual Presentation (RSVP) paradigm. Each subject participates in four experimental sessions. The training set consists of 1,654 concepts, with each concept represented by 10 unique images, and each image repeated four times, resulting in 16,540 trials per subject. The test set includes 200 unseen concepts, each represented by a single image repeated 80 times, which are averaged to obtain one trial per concept, yielding 200 test samples per subject.

EEG signals are recorded at a sampling rate of 1000 Hz and band-pass filtered between 0.1–100 Hz. The continuous signals are segmented into epochs from 0 to 1000 ms relative to stimulus onset, with baseline correction performed using the mean signal within the 200 ms pre-stimulus interval. The epoched data are then downsampled to 250 Hz for subsequent processing. To improve signal-to-noise ratio (SNR), repetitions of the same stimulus are averaged in both the training and test sets. All preprocessed EEG data are stored in `float32` format to balance storage efficiency and computational performance. For all experiments, we restrict EEG channels to those over the occipital and parietal lobes, which are closely associated with visual perception and visuospatial processing.

**THINGS-MEG.**    We further evaluate our method on the THINGS-MEG dataset, which contains magne-toencephalography (MEG) recordings from four human participants performing the same visual recognition task. The training set comprises 1,654 concepts, each associated with 12 distinct images, with one trial per image. The test set includes 200 concepts, each represented by a single image repeated 12 times, which are averaged to obtain one trial per concept. To ensure a zero-shot evaluation setting, all test concepts are entirely excluded from the training set.

MEG signals are recorded from 271 sensors at a sampling rate of 1200 Hz. Each trial includes a 500 ms stimulus presentation followed by an inter-stimulus interval of $1000\pm200$ ms. The continuous MEG recordings are segmented into epochs spanning 0 to 1000 ms after stimulus onset. A band-pass filter of 0.1–40 Hz is applied, followed by baseline correction using the mean signal from the 200 ms pre-stimulus window. The epoched signals are subsequently downsampled to 200 Hz. To enhance SNR, all repetitions corresponding to the same test image are averaged. Channel-wise z-score normalization is applied across trials. The final preprocessed MEG data are stored in `float32` format to reduce storage requirements and improve I/O efficiency during training and evaluation. For all experiments, we also restrict MEG channels to those over the occipital and parietal lobes.

## A.2    EEG Encoder Architectures

We evaluate several distinct neural network architectures for EEG signal encoding, ranging from lightweight multi-layer perceptrons to hybrid Convolutional-Transformer models. The implementation details of each model are described below.

### A.2.1    EEGProject

EEGProject serves as a lightweight baseline architecture, treating the EEG decoding task as a direct mapping problem without explicit temporal or spatial filtering layers.

- Input: The raw EEG signal $X \in \mathbb{R}^{B \times C \times T}$ is flattened into a vector of size $B \times (C \cdot T)$.

- Architecture: A Multi-Layer Perceptron (MLP) with a residual bottleneck. It consists of a linear layer, a GELU activation, dropout ($p = 0.3$), and a final LayerNorm.

### A.2.2 EEGNet

EEGNet is a compact Convolutional Neural Network specifically designed for EEG signal processing, emphasizing depthwise and separable convolutions to extract temporal and spatial features efficiently.

- Temporal Block: A 2D convolution with kernel $(1, 64)$ captures high-frequency temporal information.

- Spatial Block: A depthwise convolution with kernel $(C, 1)$ learns spatial filters across EEG channels.

- Separable Block: After average pooling $(1, 2)$, a separable convolution with kernel $(1, 16)$ integrates features followed by ELU activation and Dropout.

- Projection: The output is flattened and passed through a residual MLP block with LayerNorm to produce the final embedding.

### A.2.3 ATM (Adaptive Thinking Mapper)

The ATM model is a hybrid architecture that integrates an inverted Transformer backbone with a convolutional patch embedding module.

- Backbone (iTransformer): The backbone adopts an iTransformer-style architecture designed for multivariate time-series modeling. Given an input $X \in \mathbb{R}^{B \times C \times T}$, each time step is first mapped into a latent representation with frequency-aware positional encoding, and the resulting sequence is processed by a self-attention encoder with four attention heads and a single encoder block to capture long-range temporal dependencies across channels.

- Patch Embedding: The output of the transformer is passed to a convolutional block inspired by ShallowNet. It consists of:
    1. Temporal Convolution: Kernel $(1, 25)$, Stride $(1, 1)$.
    2. Average Pooling: Kernel $(1, 51)$, Stride $(1, 5)$.
    3. Spatial Convolution: Kernel $(C, 1)$, mapping channel interactions.

- Projection: A residual projection head maps the flattened features to the target latent dimension.

### A.2.4 EEGConformer

EEGConformer combines the local feature extraction capabilities of Convolutional Neural Networks with the global context modeling of Transformers.

- Patch Embedding: Utilizes a ShallowNet-like structure (similar to ATMS) with temporal convolution $(1, 25)$ and pooling $(1, 51)$ to downsample the signal and extract local features.

- Transformer Encoder: A stack of standard Transformer encoder blocks (default depth=2) processes the sequence of patches. Each block includes Multi-Head Self-Attention and a Feed-Forward Network with residual connections and LayerNorm.

- Projection Head: A linear projection head with BatchNorm and ELU activation maps the features to the contrastive learning space.

### A.2.5 TSConv (Time-Space Convolution)

The `TSConv` encoder focuses purely on convolutional feature extraction without attention mechanisms.

- Architecture: It employs the same convolutional front-end as the PatchEmbedding module:
    1. Conv2d $(1, 40)$ with kernel $(1, 25)$ for temporal features.
    2. AvgPool2d $(1, 51)$ with stride $(1, 5)$ for downsampling.
    3. Conv2d $(40, 40)$ with kernel $(C, 1)$ for spatial aggregation.

- Output: The features are projected via a residual linear block to the target dimension.

### A.2.6 Summary of Architecture Specifications

Table 3: Comparison of EEG Encoder Architectures and Key Hyperparameters.

| Model | Core Mechanism | Temporal Kernel | Spatial Kernel | Attention Heads |
|---|---|---|---|---|
| ATM | iTransformer + CNN | $(1, 25)$ | $(C, 1)$ | 4 |
| EEGNet | Depthwise + Separable CNN | $(1, 64)$ | $(C, 1)$ | N/A |
| EEGConformer | CNN + Transformer | $(1, 25)$ | $(C, 1)$ | 10 |
| EEGProject | MLP (Flattened) | N/A | N/A | N/A |
| TSConv | Pure CNN | $(1, 25)$ | $(C, 1)$ | N/A |

## B Results Details

### B.1 Top-1 retrieval accuracy of different vision backbones

Table 4 presents the top 1 performance of different vision backbones. Table 5 reports the intra-subject and inter-subject 200-way zero-shot retrieval accuracy on THINGS-EEG using the best-performing intermediate-layer features from different vision encoders. Overall, models with larger capacity and stronger pretraining consistently yield better decoding performance. Table 6 summarizes the decoding performance when using the final output representations of different vision encoders. Taken together, these results demonstrate that selecting appropriate intermediate representations is crucial for visual neural decoding.

Table 4: Summary of intermediate-layer alignment performance across different backbones. We report the relative depth of the best-performing intermediate layer and the final output representation. The best layer is selected based on the highest Top-1 accuracy among all probed layers. For ResNet models, the final feature is obtained by attention pooling of the last convolutional layer. For Transformer-based models, the final feature corresponds to the CLS token embedding from the last layer. Relative depth is computed as $(\ell - 1)/(L - 1)$. $\Delta$ denotes the Top-1 accuracy gain of the best intermediate layer over the final output representation.

| Model | Params | #Layers | Best $\ell$ | Best Depth (%) | Acc of Best Layer (%) | Acc of Final Output (%) | $\Delta$(%) |
|---|---|---|---|---|---|---|---|
| RN50 | 38M | 4 | 3 | 66.7 | 67.7 | 40.3 | +27.4 |
| RN101 | 56M | 4 | 3 | 66.7 | 59.2 | 36.5 | +22.7 |
| ViT-B-16 | 86M | 12 | 6 | 45.5 | 68.6 | 37.1 | +31.5 |
| ViT-H-14 | 632M | 32 | 11 | 32.3 | 76.8 | 33.5 | +43.3 |
| DINOv2 | 1.14B | 40 | 16 | 38.5 | 75.9 | 17.5 | +58.4 |
| ViT-bigG-14 | 1.84B | 48 | 27 | 55.3 | 75.2 | 33.2 | +42.0 |
| EVA-02 | 4.35B | 64 | 35 | 54.0 | 79.0 | 37.9 | +41.1 |
| InternViT | 5.54B | 46 | 28 | 60.0 | 82.6 | 55.9 | +26.7 |

Table 5: Accuracy (%) of 200-way zero-shot retrieval on THINGS-EEG using the best intermediate-layer representations from different vision encoders.

| Method | Metric | Sub1 | Sub2 | Sub3 | Sub4 | Sub5 | Sub6 | Sub7 | Sub8 | Sub9 | Sub10 | Avg. |
|---|---|---|---|---|---|---|---|---|---|---|---|---|
| **Intra-Subject: train and test on one subject** | | | | | | | | | | | | |
| RN50 | Top-1 | 62.9 | 63.3 | 68.1 | 67.7 | 58.9 | 70.6 | 65.6 | 72.1 | 66.1 | 81.9 | 67.7 |
| | Top-5 | 89.5 | 92.2 | 91.8 | 91.4 | 87.0 | 93.1 | 91.1 | 94.3 | 90.8 | 97.1 | 91.8 |
| RN101 | Top-1 | 48.4 | 56.5 | 58.9 | 60.5 | 48.2 | 62.9 | 61.7 | 66.4 | 59.0 | 69.8 | 59.2 |
| | Top-5 | 81.3 | 87.3 | 87.1 | 88.3 | 80.1 | 88.4 | 85.3 | 89.1 | 84.9 | 94.2 | 86.6 |
| ViT-B-16 | Top-1 | 63.6 | 68.3 | 69.0 | 67.3 | 61.1 | 73.0 | 68.6 | 70.4 | 66.3 | 78.3 | 68.6 |
| | Top-5 | 91.8 | 93.9 | 92.2 | 93.1 | 88.4 | 94.9 | 94.6 | 95.2 | 90.9 | 96.9 | 93.2 |
| ViT-H-14 | Top-1 | 69.4 | 78.9 | 78.6 | 73.1 | 68.7 | 81.1 | 78.4 | 80.1 | 75.0 | 85.0 | 76.8 |
| | Top-5 | 92.2 | 97.1 | 96.4 | 96.4 | 94.2 | 97.5 | 96.1 | 98.4 | 94.7 | 97.4 | 96.0 |
| DINOv2 | Top-1 | 67.7 | 77.9 | 76.0 | 72.7 | 69.7 | 81.9 | 74.1 | 79.1 | 75.6 | 84.7 | 75.9 |
| | Top-5 | 92.2 | 95.5 | 96.2 | 95.0 | 92.2 | 98.2 | 96.2 | 98.5 | 95.2 | 98.3 | 95.8 |
| ViT-bigG-14 | Top-1 | 65.3 | 75.1 | 74.4 | 75.3 | 67.7 | 82.7 | 74.3 | 79.8 | 70.9 | 86.8 | 75.2 |
| | Top-5 | 90.8 | 95.0 | 95.5 | 96.4 | 92.7 | 97.0 | 94.8 | 98.3 | 92.8 | 95.9 | 94.9 |
| EVA-02 | Top-1 | 70.8 | 81.9 | 80.7 | 77.6 | 74.3 | 84.2 | 76.8 | 82.7 | 75.8 | 85.2 | 79.0 |
| | Top-5 | 92.8 | 97.7 | 95.9 | 98.2 | 94.5 | 97.4 | 95.6 | 98.6 | 96.0 | 98.5 | 96.5 |
| InternViT | Top-1 | 75.0 | 87.5 | 83.2 | 79.5 | 74.6 | 89.9 | 78.5 | 86.9 | 81.3 | 89.3 | 82.6 |
| | Top-5 | 94.3 | 98.9 | 98.2 | 96.1 | 96.4 | 99.3 | 97.3 | 99.4 | 97.8 | 99.1 | 97.7 |
| **Inter-Subject: leave one subject out for test** | | | | | | | | | | | | |
| RN50 | Top-1 | 16.7 | 24.4 | 7.6 | 18.1 | 13.4 | 14.0 | 14.5 | 15.7 | 17.7 | 22.2 | 16.4 |
| | Top-5 | 42.0 | 51.1 | 27.3 | 43.4 | 41.4 | 37.5 | 36.7 | 38.8 | 44.4 | 52.1 | 41.5 |
| RN101 | Top-1 | 16.2 | 22.0 | 8.3 | 15.9 | 13.0 | 13.4 | 11.1 | 13.3 | 16.7 | 21.8 | 15.2 |
| | Top-5 | 38.3 | 49.2 | 28.1 | 42.0 | 37.4 | 38.1 | 33.3 | 36.7 | 43.3 | 51.8 | 39.8 |
| ViT-B-16 | Top-1 | 18.3 | 23.7 | 7.4 | 17.8 | 14.1 | 12.1 | 14.4 | 15.0 | 17.6 | 22.7 | 16.3 |
| | Top-5 | 42.8 | 52.1 | 26.3 | 45.1 | 40.0 | 39.3 | 38.4 | 36.1 | 46.1 | 54.7 | 42.1 |
| ViT-H-14 | Top-1 | 21.6 | 28.5 | 10.3 | 20.2 | 16.5 | 16.0 | 16.9 | 16.1 | 18.9 | 24.5 | 19.0 |
| | Top-5 | 48.4 | 59.6 | 26.3 | 45.4 | 43.6 | 40.4 | 40.5 | 36.6 | 46.6 | 55.1 | 44.2 |
| DINOv2 | Top-1 | 23.2 | 26.7 | 11.2 | 18.8 | 18.0 | 16.6 | 16.8 | 13.2 | 18.9 | 23.3 | 18.7 |
| | Top-5 | 53.4 | 60.6 | 29.0 | 45.4 | 44.8 | 43.5 | 45.4 | 36.8 | 49.1 | 56.0 | 46.4 |
| ViT-bigG-14 | Top-1 | 19.0 | 30.0 | 16.0 | 19.0 | 17.0 | 20.0 | 16.5 | 14.0 | 22.5 | 33.5 | 20.8 |
| | Top-5 | 56.5 | 57.5 | 31.0 | 44.0 | 41.5 | 48.5 | 38.0 | 40.0 | 56.0 | 57.5 | 47.0 |
| EVA-02 | Top-1 | 20.8 | 26.5 | 11.5 | 18.8 | 18.9 | 18.8 | 14.5 | 17.8 | 20.5 | 24.8 | 19.3 |
| | Top-5 | 50.9 | 57.2 | 29.0 | 45.2 | 44.7 | 46.6 | 41.2 | 40.2 | 53.6 | 53.8 | 46.2 |
| InternViT | Top-1 | 23.5 | 30.6 | 10.0 | 19.5 | 18.1 | 22.7 | 18.6 | 17.3 | 23.0 | 34.4 | 21.8 |
| | Top-5 | 53.2 | 60.4 | 28.1 | 48.3 | 45.2 | 49.8 | 46.0 | 46.1 | 54.8 | 62.0 | 49.4 |

Table 6: Accuracy (%) of 200-way zero-shot retrieval on THINGS-EEG using the final representations from different vision encoders.

| Method | Metric | Sub1 | Sub2 | Sub3 | Sub4 | Sub5 | Sub6 | Sub7 | Sub8 | Sub9 | Sub10 | Avg. |
|--------|--------|------|------|------|------|------|------|------|------|------|-------|------|
| **Intra-Subject: train and test on one subject** | | | | | | | | | | | | |
| RN50 | Top-1 | 26.3 | 38.9 | 42.1 | 40.7 | 32.2 | 46.8 | 39.0 | 47.9 | 36.0 | 53.0 | 40.3 |
|      | Top-5 | 59.9 | 72.1 | 78.5 | 75.5 | 65.5 | 77.5 | 70.1 | 79.9 | 70.6 | 85.7 | 73.5 |
| RN101 | Top-1 | 28.9 | 35.3 | 35.9 | 34.8 | 30.2 | 38.8 | 35.8 | 43.1 | 33.9 | 48.5 | 36.5 |
|       | Top-5 | 59.8 | 66.6 | 71.8 | 72.0 | 60.6 | 72.2 | 69.9 | 75.7 | 69.4 | 82.2 | 70.0 |
| ViT-B-16 | Top-1 | 28.5 | 35.0 | 39.8 | 36.2 | 29.0 | 40.8 | 36.2 | 41.1 | 35.2 | 49.1 | 37.1 |
|          | Top-5 | 58.8 | 68.7 | 72.8 | 71.9 | 59.1 | 72.5 | 70.3 | 76.2 | 66.9 | 83.2 | 70.1 |
| ViT-H-14 | Top-1 | 23.6 | 29.0 | 38.9 | 35.9 | 26.0 | 36.1 | 29.2 | 40.6 | 31.9 | 44.2 | 33.5 |
|          | Top-5 | 52.7 | 63.8 | 67.1 | 65.0 | 55.7 | 69.1 | 63.7 | 71.8 | 64.2 | 76.4 | 65.0 |
| DINOv2 | Top-1 | 11.9 | 16.0 | 18.6 | 18.4 | 12.0 | 17.3 | 17.4 | 23.0 | 17.2 | 23.2 | 17.5 |
|        | Top-5 | 30.2 | 40.1 | 46.5 | 42.5 | 32.1 | 41.3 | 40.7 | 47.8 | 39.6 | 47.7 | 40.9 |
| ViT-bigG-14 | Top-1 | 20.7 | 31.1 | 36.8 | 30.3 | 27.9 | 38.4 | 32.9 | 40.0 | 30.0 | 43.8 | 33.2 |
|             | Top-5 | 49.4 | 65.5 | 70.2 | 64.2 | 51.9 | 75.4 | 63.5 | 71.6 | 66.8 | 78.1 | 65.7 |
| EVA-02 | Top-1 | 28.7 | 36.6 | 41.1 | 36.6 | 28.6 | 40.8 | 38.6 | 42.6 | 36.0 | 49.8 | 37.9 |
|        | Top-5 | 55.9 | 67.7 | 73.1 | 71.0 | 60.8 | 76.2 | 69.5 | 76.0 | 64.6 | 81.9 | 69.7 |
| InternViT | Top-1 | 44.1 | 54.4 | 56.3 | 57.4 | 46.4 | 59.0 | 52.5 | 64.3 | 53.8 | 71.2 | 55.9 |
|           | Top-5 | 76.5 | 82.5 | 88.5 | 87.0 | 76.9 | 90.4 | 85.0 | 90.6 | 84.3 | 94.9 | 85.7 |
| **Inter-Subject: leave one subject out for test** | | | | | | | | | | | | |
| RN50 | Top-1 | 9.6 | 12.7 | 4.6 | 11.7 | 7.7 | 7.7 | 9.2 | 9.6 | 8.7 | 11.3 | 9.3 |
|      | Top-5 | 29.8 | 35.2 | 19.8 | 30.5 | 24.6 | 26.7 | 26.6 | 27.1 | 29.6 | 33.6 | 28.4 |
| RN101 | Top-1 | 9.3 | 12.9 | 3.3 | 11.1 | 6.7 | 6.1 | 7.3 | 8.0 | 4.2 | 12.7 | 8.1 |
|       | Top-5 | 25.4 | 35.1 | 17.1 | 30.8 | 20.0 | 23.9 | 24.0 | 22.4 | 20.9 | 35.0 | 25.5 |
| ViT-B-16 | Top-1 | 9.5 | 12.6 | 3.6 | 11.4 | 6.8 | 7.8 | 8.3 | 6.9 | 6.2 | 14.4 | 8.7 |
|          | Top-5 | 26.4 | 32.5 | 15.6 | 28.4 | 23.0 | 24.1 | 24.6 | 26.0 | 23.8 | 37.8 | 26.2 |
| ViT-H-14 | Top-1 | 8.0 | 10.2 | 5.2 | 12.0 | 5.4 | 6.2 | 4.7 | 7.4 | 7.3 | 11.3 | 7.8 |
|          | Top-5 | 23.9 | 26.2 | 15.6 | 31.2 | 17.2 | 22.1 | 23.0 | 20.9 | 20.8 | 34.2 | 23.5 |
| DINOv2 | Top-1 | 5.7 | 8.8 | 5.3 | 7.0 | 4.2 | 6.0 | 5.0 | 6.1 | 5.7 | 7.3 | 6.1 |
|        | Top-5 | 20.1 | 24.5 | 13.9 | 20.9 | 16.2 | 18.4 | 16.1 | 18.5 | 17.7 | 25.4 | 19.2 |
| ViT-bigG-14 | Top-1 | 8.0 | 10.3 | 5.3 | 7.8 | 5.5 | 10.0 | 7.0 | 7.0 | 7.0 | 8.5 | 7.6 |
|             | Top-5 | 24.8 | 29.0 | 18.8 | 28.5 | 22.0 | 25.5 | 25.0 | 24.5 | 23.0 | 32.3 | 25.3 |
| EVA-02 | Top-1 | 9.9 | 11.6 | 4.5 | 10.6 | 5.7 | 8.1 | 6.2 | 7.6 | 8.2 | 12.1 | 8.5 |
|        | Top-5 | 26.9 | 30.7 | 17.8 | 32.7 | 24.2 | 25.6 | 24.7 | 28.6 | 26.9 | 34.9 | 27.3 |
| InternViT | Top-1 | 16.0 | 19.7 | 8.0 | 14.2 | 10.0 | 10.4 | 10.4 | 12.6 | 15.2 | 19.6 | 13.6 |
|           | Top-5 | 40.2 | 45.7 | 25.0 | 40.3 | 31.5 | 37.2 | 30.3 | 33.3 | 42.2 | 45.7 | 37.2 |

### B.2 Assessing Robustness to Validation Splits

For fair comparison with prior baselines, the main experiments follow the standard protocol and train on the full training set without a validation split. We additionally conduct a validation-split analysis to verify that this choice does not affect our conclusions.

From the 1,654 training concepts, we randomly hold out 74 concepts, corresponding to approximately 5% of the training concepts, and train on the remaining concepts, corresponding to $1580 \times 10$ trial-level samples for THINGS-EEG and $1580 \times 12$ for THINGS-MEG. After each epoch, validation loss is computed on the 74 held-out concepts after averaging trials within each concept, yielding 74 concept-level validation samples. The best checkpoint is selected by the lowest validation loss, and the official test set is evaluated only once.

As shown in Fig. 5, introducing the validation split produces highly similar layer-wise trends to the original setting. In both cases, performance increases from shallow layers, peaks at intermediate depths, and decreases toward the final output layer. The absolute accuracies show only minor fluctuations, and the best-performing layers remain intermediate rather than final layers. Therefore, using the full training set in the main experiments does not materially affect the paper's conclusion that intermediate representations provide more effective alignment targets than final-layer representations.

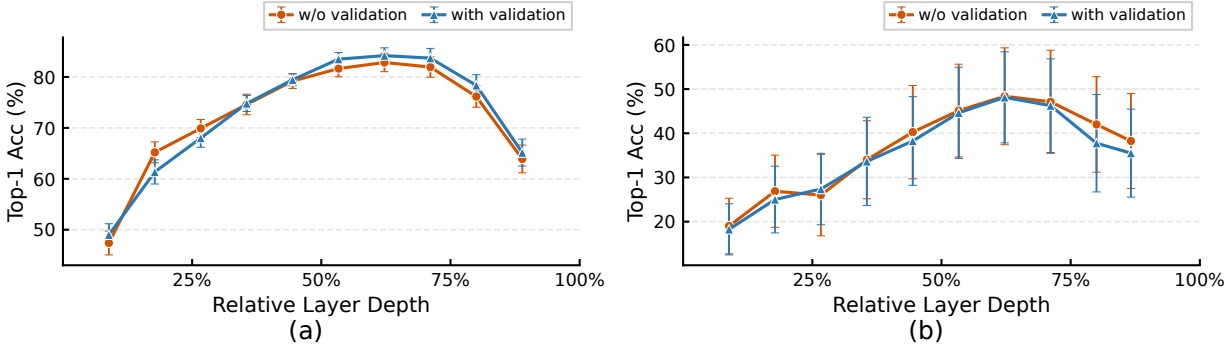

Figure 5: Top-1 retrieval accuracy across InternViT layers (relative depth) for InternViT avg-pool features on (a) THINGS-EEG2 intra-subject (10 subjects) and (b) THINGS-MEG intra-subject (4 subjects). Two model-selection strategies are compared: w/o validation (orange) and with validation (blue). Error bars: mean ± SEM.

### B.3 Evaluating Encoder Efficacy via Architectural Comparison

To assess the impact of encoder architecture on cross-modal alignment performance, we examine combinations of five EEG encoders and eight vision encoders (See details in Appendix A.2). Our empirical evaluation highlights the superior efficacy of EEGProject. Despite its architectural simplicity, it outperforms more complex baselines (e.g., EEGNet, ATM).

As shown in Figure 9, the lightweight EEGProject model consistently attains the highest performance across all vision backbones, achieving the highest average accuracy of 73.1%. This outcome aligns with the intrinsic nature of EEG data, which is characterized by a low signal-to-noise ratio and data scarcity. In this regime, complex architectures struggle to generalize. For instance, despite its high capacity, the widely used EEGNet averages only 53.8% accuracy, lagging behind our simpler MLP-based approach by nearly 20%.

Conversely, EEGProject imposes a tighter information bottleneck that forces the encoder to distill the most discriminative features while suppressing task-irrelevant artifacts. This results in representations that are better aligned with the visual embedding space. These findings suggest that, for neural visual decoding, a simple architecture can be more effective when paired with appropriately chosen visual representations.

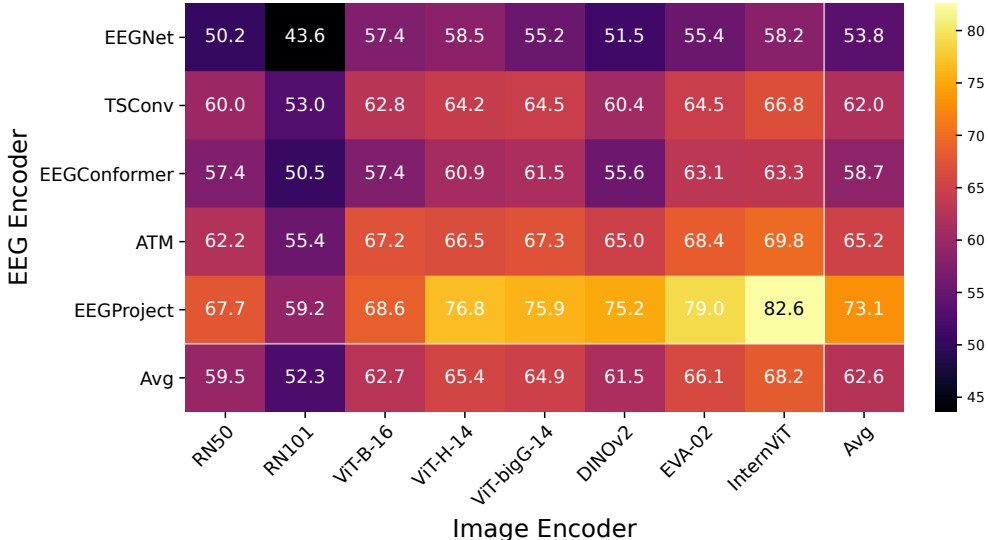

Figure 6: Top-1 accuracy (%) for various encoders architectures on THINGS-EEG.

## B.4  Effect of Restricting EEG to Occipital–Parietal Channels

**Setup.**  Our main experiments use the 17-channel occipital–parietal (OP) montage from the THINGS-EEG2. To quantify the impact of this choice, we conducted the same experiment with five region-based subsets (frontal F: 26 ch, central C: 21 ch, parietal P: 25 ch, occipital O: 8 ch, temporal T: 10 ch) and the full 63-channel montage, on four backbones spanning CNN and ViT architectures (RN50, RN101, ViT-H-14, InternViT-6B). All other settings match the main experiments.

As shown in Fig. 7: (i) the F, C, and T regions carry essentially no decodable signal, peaking at less than 12% Top-1 across all backbones, and including them via the full 63-channel input reduces peak accuracy by 12–18 percentage points relative to OP (e.g., InternViT-6B drops from 82.2% to 70.9%); (ii) the optimal alignment layer is invariant to channel selection, with the inverted-U peaking at the same relative depth across all six configurations on every backbone (∼75% for the two CNNs, 34% for ViT-H-14, 62% for InternViT), so no alternative layer becomes preferred for a different scalp region; Within the present montage, however, the OP restriction preserves essentially all decodable object-category information and does not affect the identification of the most EEG-aligned layer.

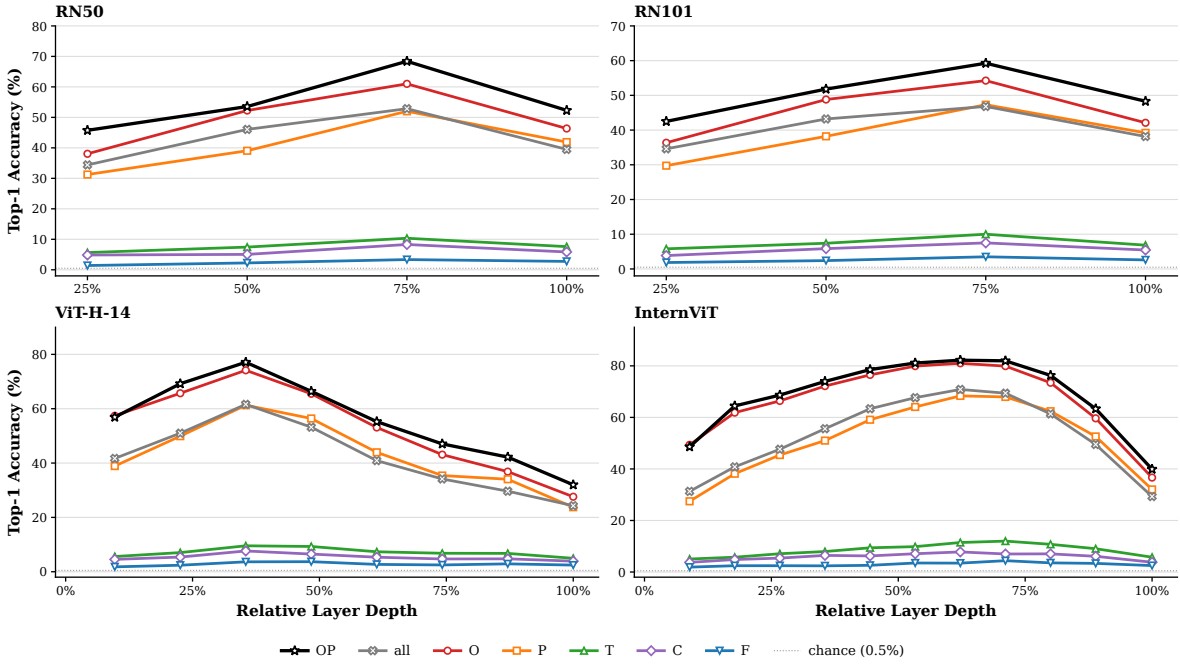

Figure 7:    Layer-wise EEG-image retrieval accuracy across visual-encoder backbones and EEG channel groups. Top-1 accuracy on THINGS-EEG2 (mean ± std over 10 subjects, intra-subject training, 30 epochs) plotted against relative layer depth for four pretrained visual encoders: RN50 (top-left), RN101 (top-right), ViT-H-14 (bottom-left), and InternViT-6B (bottom-right). For each backbone we compare seven EEG-channel configurations: the OP electrodes used in the main paper (17 channels covering occipital and parietal-occipital sites, black stars); the all 63-channel montage (gray crosses); and the five region-based subgroups defined by whether the channel name contains the letter O (occipital, 8 ch), P (parietal, 25 ch), T (temporal, 10 ch), C (central, 21 ch), or F (frontal, 26 ch). The dotted line marks chance accuracy (0.5%). The x-axis shows each layer's transformer/CNN block index normalized by the backbone's total number of blocks.

## B.5 Subject-Level Consistency of the Best-Aligned Feature Layer

We examined whether the best-aligned feature layer varies across subjects. For each image encoder, we computed the Top-1 retrieval accuracy at each feature layer separately for each subject, and identified the layer with the highest accuracy for that subject. Figure 8 shows the resulting subject-wise layer-alignment profiles as a function of relative feature depth.

Overall, the best-aligned layer was largely consistent within each model. For RN50, all subjects peaked at the same intermediate layer, and ViT-H-14 showed the same pattern, with all subjects selecting the same relative feature depth. RN101 also showed minimal variation, with most subjects peaking at the same layer. In contrast, larger ViT-family models such as DINOv2, ViT-bigG-14, EVA-02, and InternViT showed modest subject-to-subject variation, but the selected layers remained concentrated around nearby intermediate depths.

These results suggest that subject differences do not strongly alter the preferred representational stage. Instead, subject variability appears mainly as a difference in overall decoding accuracy, while the depth at which visual features best align with EEG responses is primarily determined by the image encoder and its representational hierarchy. Thus, although individual subjects can show small shifts in the selected layer, the dominant pattern is a stable model-specific intermediate-layer alignment.

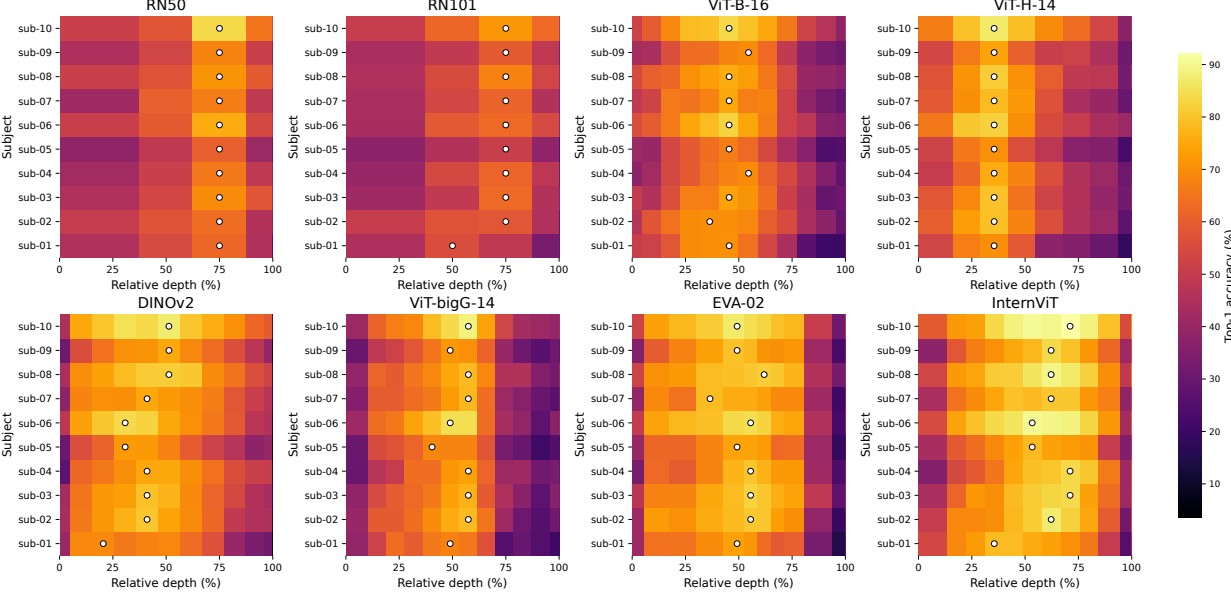

Figure 8: Subject-wise layer alignment analysis. Heatmaps show Top-1 accuracy across relative feature depth for each subject and image encoder. White markers indicate the best-aligned layer for each subject. The optimal layer is highly consistent for some models, such as RN50 and ViT-H-14, while larger ViT-family models show modest subject-to-subject variation around nearby intermediate layers. Overall, the alignment peak is primarily model- and layer-dependent, with subject variation mainly affecting accuracy magnitude rather than producing large shifts in the preferred feature depth.

### B.6 Robustness of Scaling Analysis to Parameter-Count Definition

In the main text, we use total backbone parameters as the scaling variable because they reflect the capacity of the pretrained model family. Here, we repeat the analysis using effective parameters, defined as the number of parameters up to the selected intermediate layer. This stricter definition measures the capacity directly involved in computing the visual target. The results remain consistent with the main analysis: intermediate-layer alignment shows a significant positive scaling trend, whereas final-layer alignment does not.

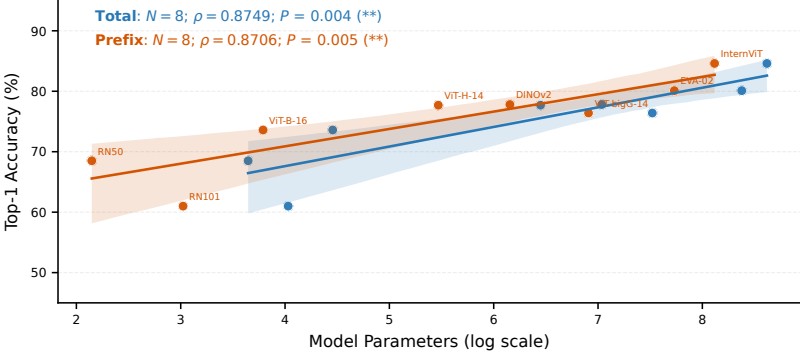

Figure 9: Scaling analysis. Linear regression analysis reveals the relationship between model scale and Top-1 accuracy. Model scale is measured using both the total number of parameters and the prefix parameter count up to the selected intermediate layer, both in ln scale. Both parameter-count definitions show significant positive correlations with Top-1 accuracy (total: $\rho = 0.8749$, $p = 0.004$; prefix: $\rho = 0.8706$, $p = 0.005$), indicating that the scaling trend is robust to the parameter-count definition.

### B.7 Analysis of Layer-wise Training Dynamics

To validate the impact of feature granularity, we analyze the training and testing loss curves across different layers of the InternViT encoder, as shown in Figure 10. The optimization landscapes reveal three distinct behaviors:

- Shallow Layers (e.g., Layers 4–12): These layers converge slowly and plateau at a higher testing loss, indicating that low-level features alone lack sufficient semantic discriminability for effective retrieval.

- Optimal Intermediate Layers (e.g., Layers 24–28): The most favorable dynamics emerge in the middle layers. Specifically, Layer 28 achieves the lowest testing loss among all candidates and maintains a minimal gap between training and testing curves. This tight generalization bound confirms that intermediate representations possess the superior granularity—balancing semantic abstraction with necessary texture details—to align with neural signals.

- Deep Layers (e.g., Layer 40–Final): Deeper layers exhibit signs of severe overfitting. While the training loss drops rapidly to near-zero, the test loss (orange) remains significantly high. This large generalization gap supports our "Granularity Mismatch" hypothesis: the highly compressed, invariant features at the end of large models are too abstract for the noisy neural signals to predict effectively.

In conclusion, the loss analysis corroborates our quantitative results: simply scaling up model depth does not guarantee better decoding. Instead, selecting the granularity-matched intermediate layer is crucial for preventing overfitting and achieving robust alignment.

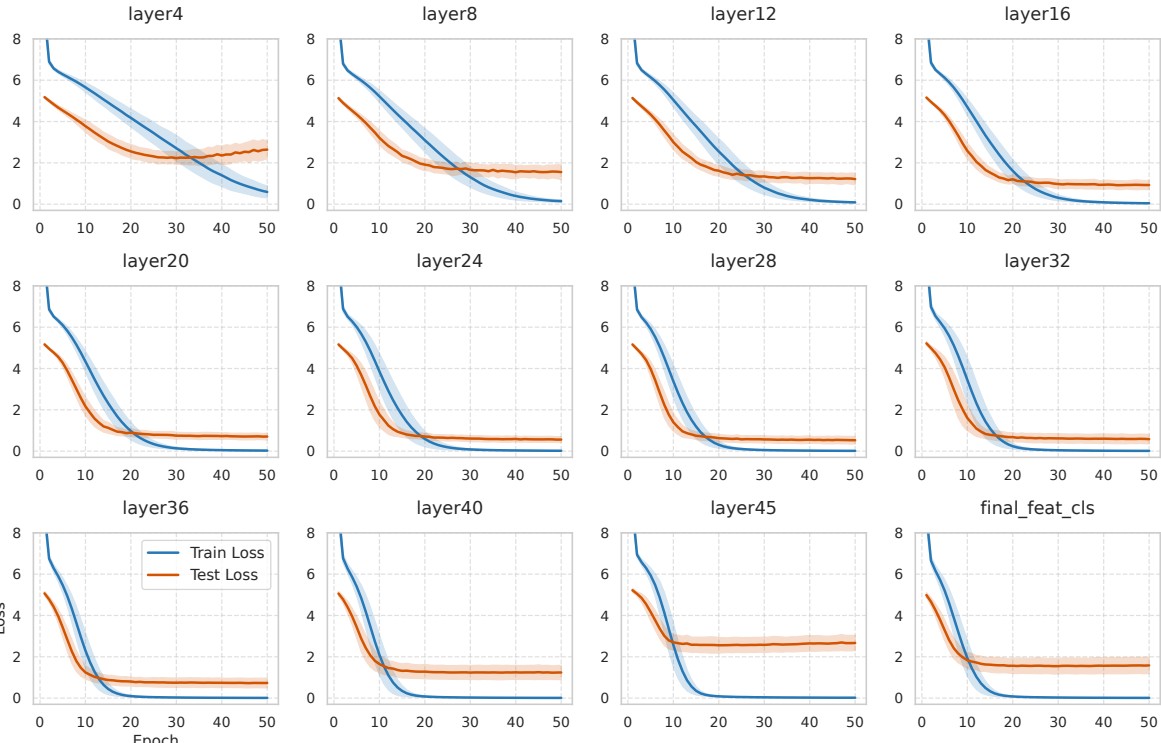

Figure 10: Layer-wise training dynamics on InternViT.

## B.8 UMAP Analysis

As shown in Figure 11, we can see that the two modalities gradually transition from being well separated at early layers to achieving the strongest mixing and alignment at intermediate layers (e.g., Layers 20–32), indicating that representations at these depths are most consistent with the intrinsic structure of neural signals. However, as the network deepens toward the final output layer, the feature distributions become separated again, further suggesting a granularity mismatch between the model's highly abstract semantic representations and the human visual representations that retain rich fine-grained details.

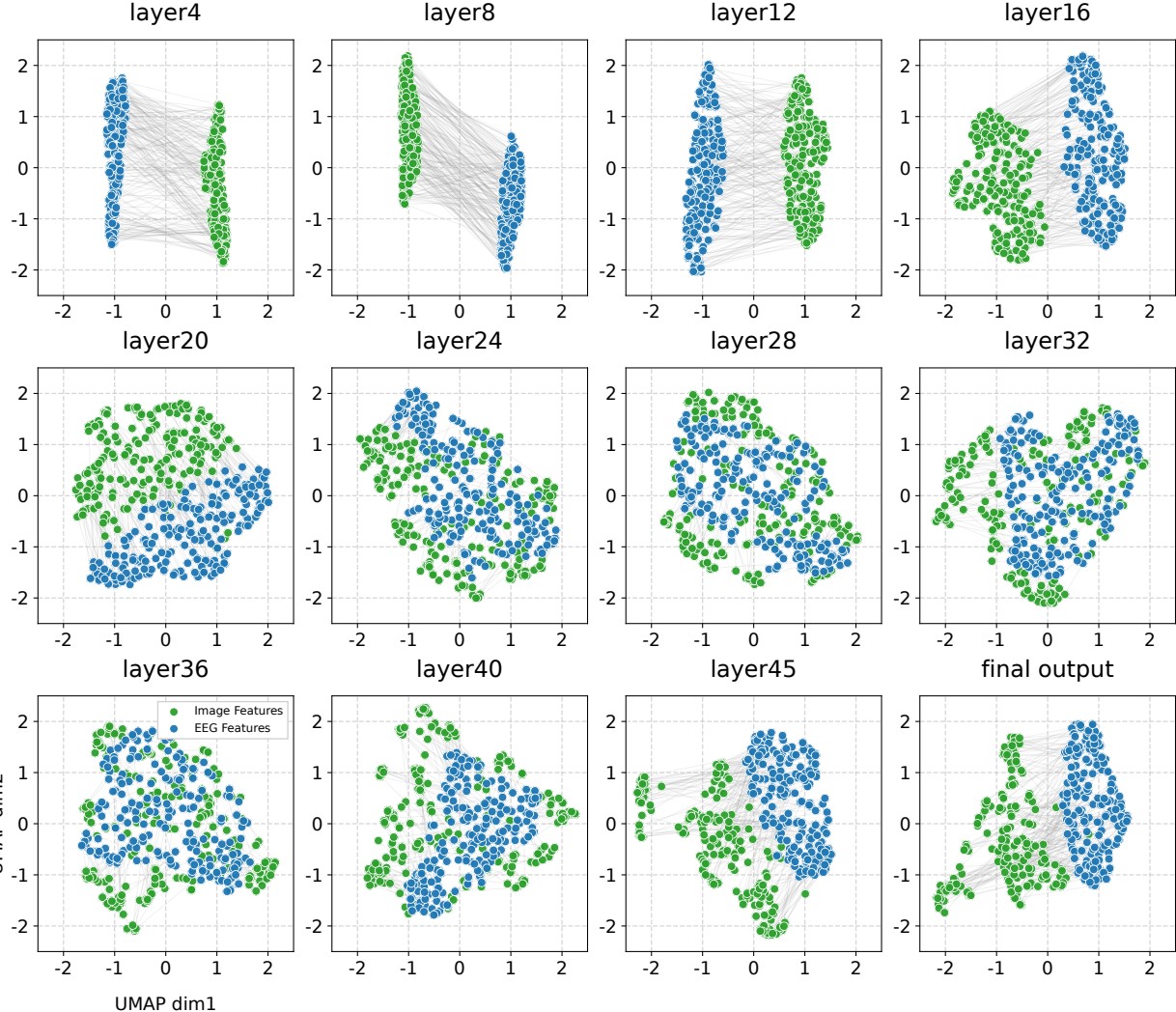

Figure 11: UMAP visualization of cross-modal alignment on THINGS-EEG (Subject 7). Illustration of alignment between image and EEG features on the test set of Subject 7 from the THINGS-EEG dataset, consisting of 200 samples. Image features are extracted from InternViT, while EEG features are obtained from the neural encoder. Each gray line connects a paired image–EEG sample.

### B.9 Validating Semantic Collapse via Linear Projection Ablation

We conduct an ablation study on the linear projection to examine its role in alignment. As shown in Figure 12, we observe a clear contrast in the effectiveness of linear projection across different representation depths. When applied to the final-layer visual embeddings, introducing a learnable linear projector yields only marginal performance improvements. In contrast, linear projection substantially improves alignment performance when applied to intermediate representations. The most pronounced gain is observed for ResNet-50, where Top-1 accuracy increases from 28.8% to 67.7%, corresponding to an absolute improvement of nearly 40%. The limited benefit at the final layer suggests that these representations have undergone severe semantic collapse, such that a simple linear transformation is insufficient to recover the structural information necessary for effective alignment. Conversely, the significant improvements observed at intermediate layers indicate that these representations retain richer structural fidelity. This richness allows the linear layer to function as a selective filter, identifying the specific visual primitives that best match the granularity of neural signals while discarding task-irrelevant noise.

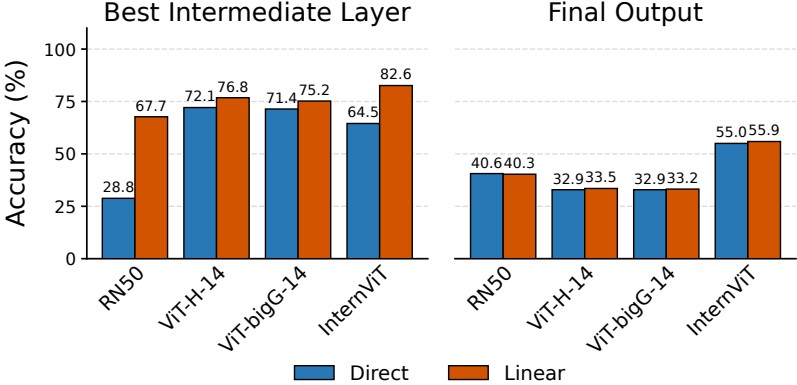

Figure 12: Top 1 accuracy comparison across different projector types on THINGS-EEG. Results are shown for the best intermediate layer and the final output layer.

## B.10 Retrieval Case

We present more top-5 retrieval results on THINGS-EEG dataset in Figure 13, 14, 15, and 16, respectively. Our results indicate that fine-grained differences exist not only between different layers of the vision encoder, but also across subjects.

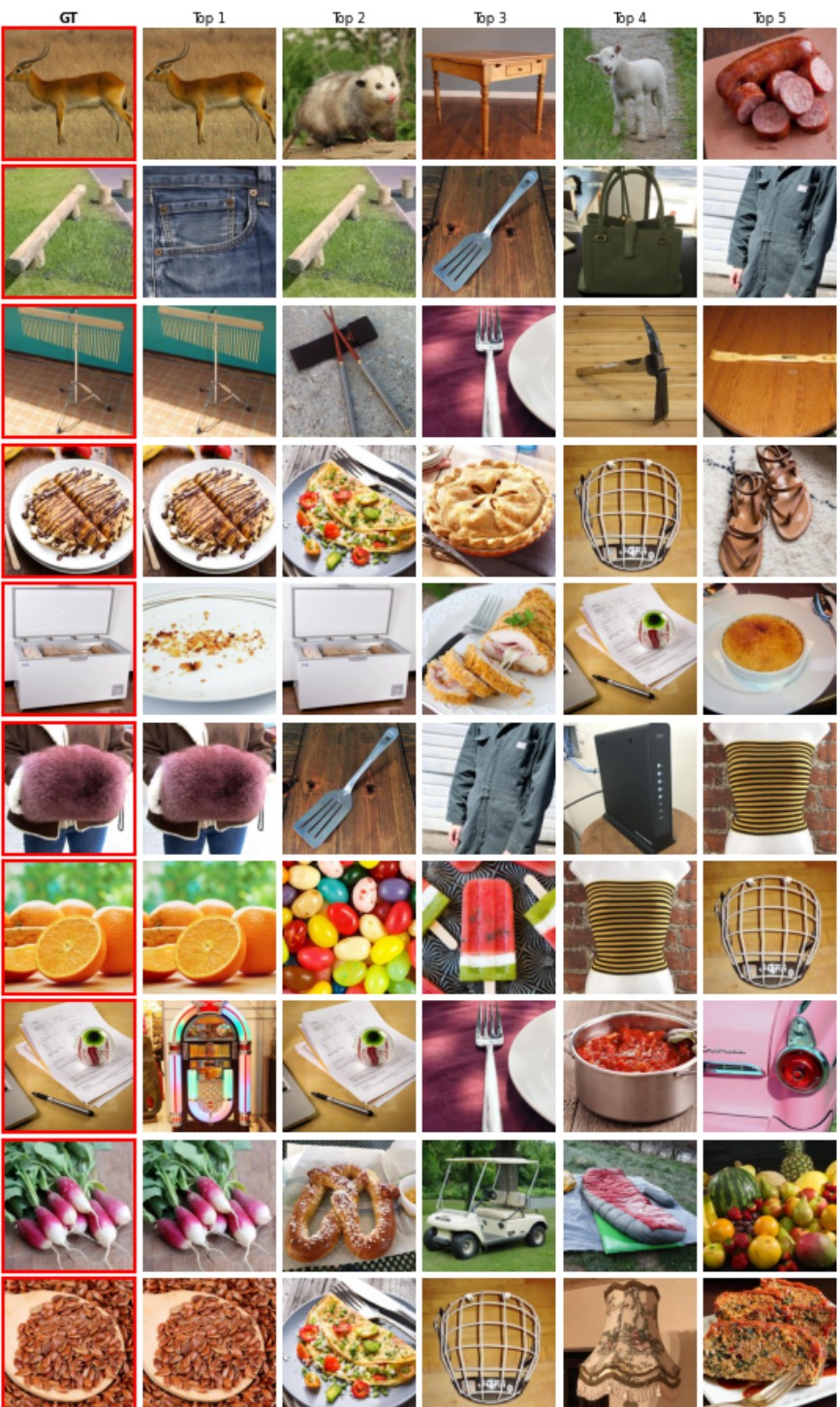

Figure 13: More top-5 retrieved samples of Subject 7 based on the best intermediate-layer embeddings.

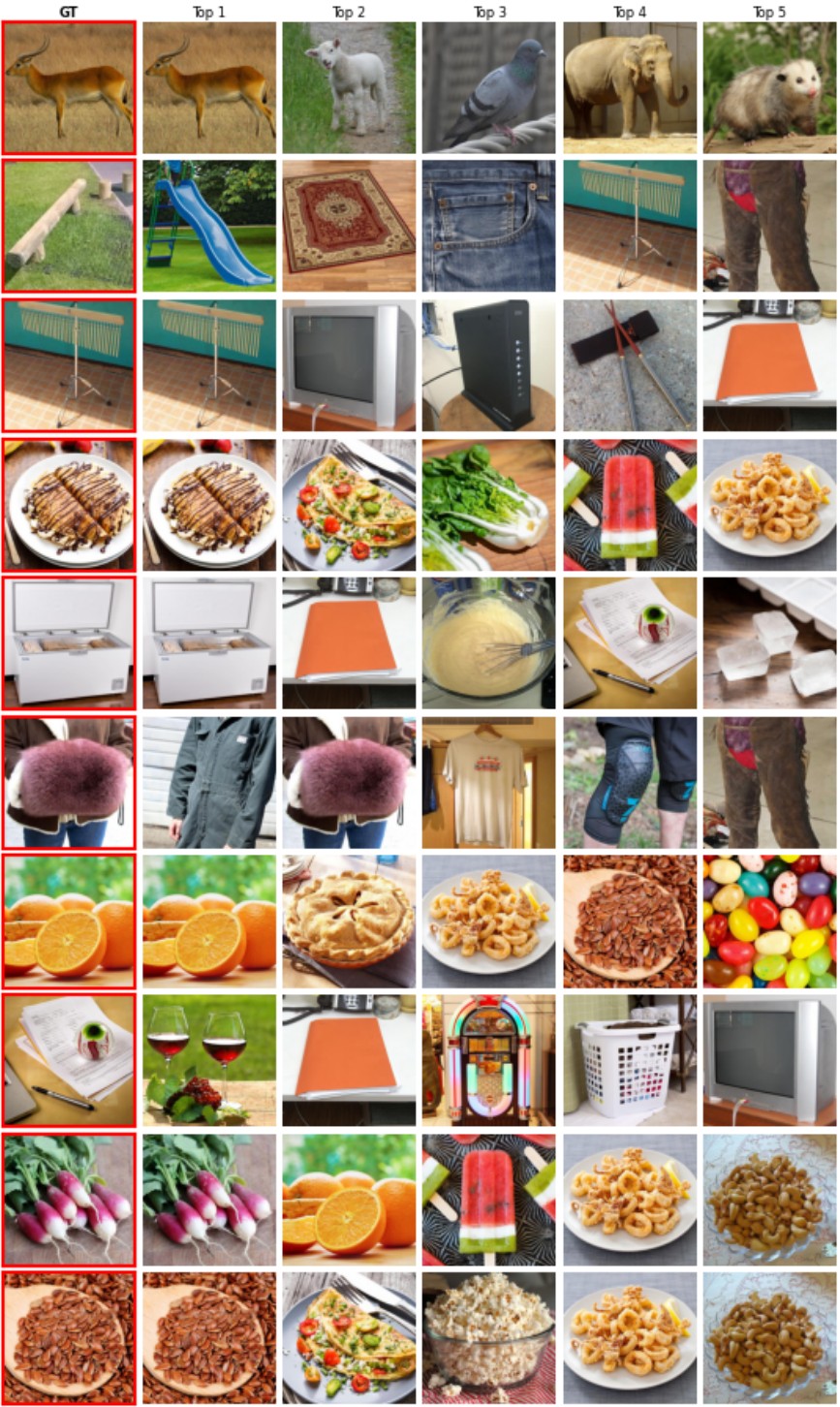

Figure 14: More top-5 retrieved samples of Subject 7 based on the final output embedding.

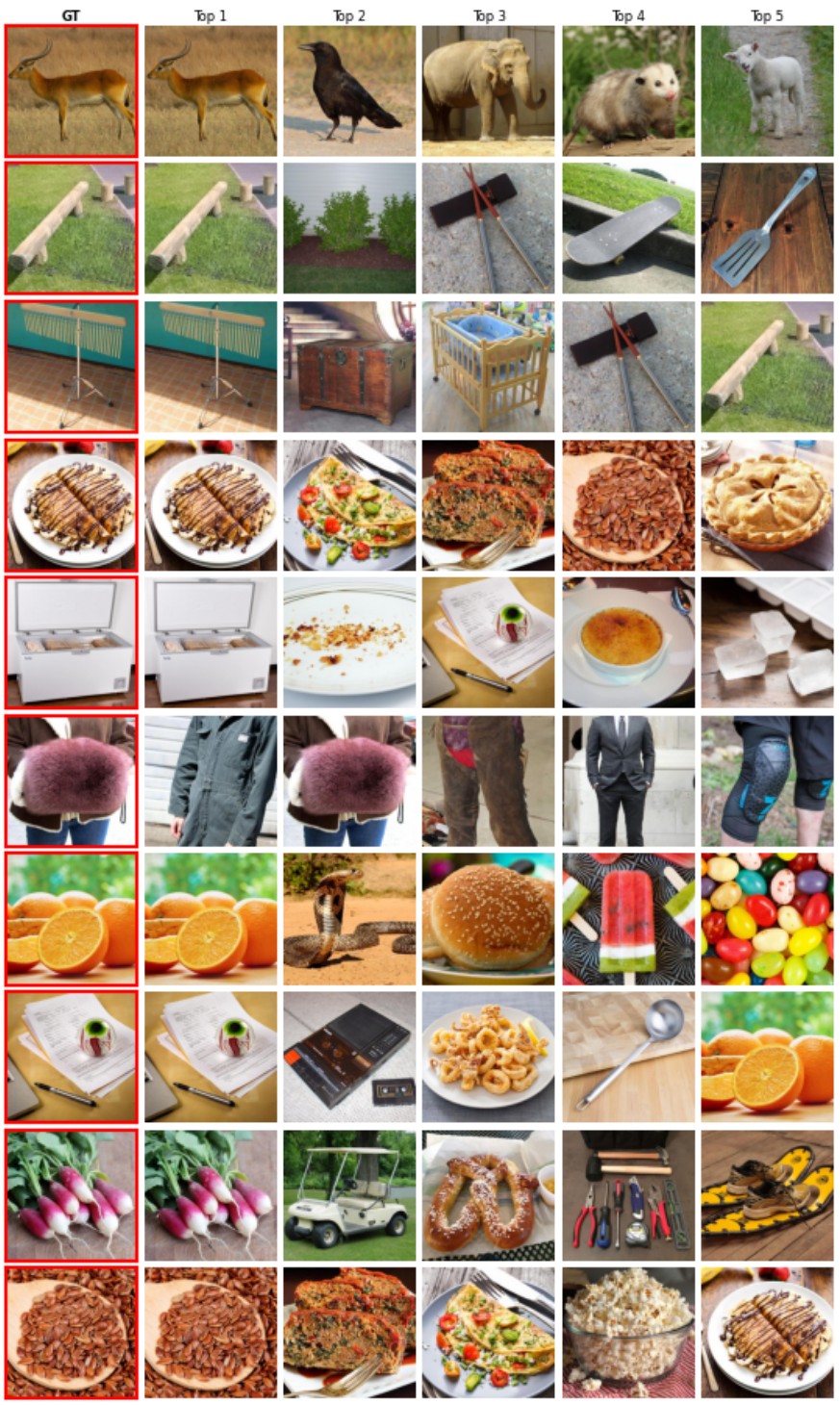

Figure 15: More top-5 retrieved samples of Subject 10 based on the best intermediate-layer embeddings.

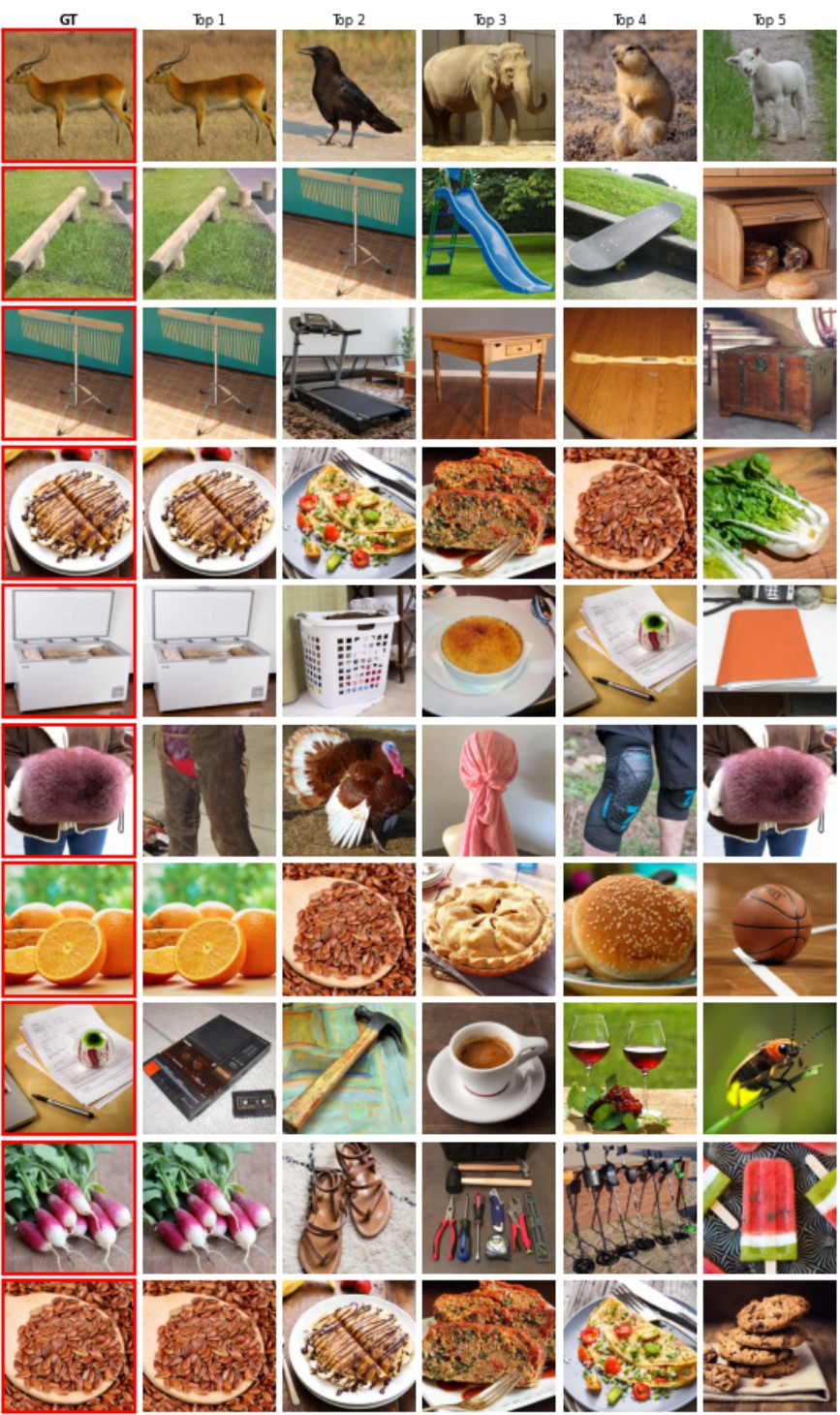

Figure 16: More top-5 retrieved samples of Subject 10 based on the final output embedding.

