# OpenReview forum: "Deep Models, Shallow Alignment: Uncovering the Granularity Mismatch in Neural Decoding"
_TMLR — Under review for TMLR_

### Review · Reviewer_9UHe · 2026-04-22

**Summary Of Contributions:**

The paper suggests a method to maximize decoding of visual images from EEG/MEG signals. The authors suggest the use of intermediate layers instead of the last layer of vision DNN as representational targets for decoding neural data.

**Audience:**

No

**Audience Explanation:**

Most of the ML community approach this problem for being interested in the topic of brain decoding. However, decoding as a tool of neuroscience should serve to understand the brain processing which is not the focus of the paper in current form.
The BCI angle for this is not exactly important as BCI is more interested in decoding of action (for people with lack of mobility) and encoding of sensory signal (for people with a certain sensory impairment, like vision). This work focuses on decoding of sensory signals.

**Claims And Evidence:**

No

**Claims Explanation:**

The problem of alignment between brain and DNN models has been extensively studies from fMRI perspective where it has been shown that the alignment between features from various regions from the visual hierarchy will be homologous with those from DNNs (check out  Horikawa et al. 2017 Nature Communications). So my first intuition to this would be that since IT and early visual areas are located in deeper regions than what EEG/MEG can capture, the data comes primarily from intermediate visual areas. This means that the claim of granularity match is not necessarily generalizable. If anything, I would be surprised if the EEG community has not already investigated this.

**Requested Changes:**

A deeper neuroscience-focused literature search should help refocus the paper and understand what is known and what are the gaps in literature.

Issues with the technical details:
The authors should explain how the algorithm selects the optimal layer. It appears so far that the choice is made aposteriori.
It is unclear where the UMAP projection comes from. As in what is the input feature? Is it a concatenation of the decoded neural and DNN features?

---

> ### Author Response · Authors · 2026-05-13
> **Response to Reviewer 9UHe**
>
> We thank the reviewer for the constructive feedback. We clarify that our work is a methodological study on the choice of visual alignment target in EEG/MEG-based contrastive decoding, not a neuroscience claim. We have tightened the framing, expanded the related work, and clarified the layer sweep and UMAP procedures accordingly. Detailed responses follow.
>
> ---
>
> ### Clarification on the evidence supporting our claim
>
> We would like to clarify that our paper does not aim to show that EEG/MEG captures signals from a particular visual region such as IT, nor do we claim a precise anatomical mapping between EEG/MEG signals and stages of the visual hierarchy. Our claim is instead about the choice of visual alignment target in neural visual decoding. EEG/MEG signals contain visual information at multiple levels of granularity, including both lower-level visual structure and higher-level semantic information. Therefore, for alignment-based decoding, the visual target should ideally preserve both types of information. In the current EEG/MEG-based contrastive decoding paradigm, prior methods use the final-layer embedding of a pretrained vision model as the default representational target. Our work revisits this design choice and shows that intermediate representations can provide a substantially more effective supervision signal.
>
> We agree that brain-DNN alignment has been extensively studied, particularly in fMRI work on hierarchical visual representations. We have therefore expanded the related work section to include additional EEG/MEG studies on DNN-neural alignment. We have also revised the manuscript, mainly in the **Introduction** and **Conclusion**, to make this scope clearer and softened several statements that may have unintentionally suggested an overclaim.
>
> ---
>
> ### Clarification on the paper's relevance to the TMLR audience
>
> We believe the findings are relevant to TMLR readers because the paper provides a systematic characterization of how pretrained vision-model representations at different depths align with EEG/MEG signals in neural decoding. In particular, our results show that intermediate layers provide stronger alignment targets than final-layer embeddings, suggesting that the representation used for alignment is a central design.
>
> ---
>
> ### Requested Change 1: *A deeper neuroscience-focused literature search should help refocus the paper and understand what is known and what are the gaps in literature*
>
> We would like to respectfully clarify that our paper is not a neuroscience contribution but a methodological study on the choice of visual alignment target in EEG/MEG-based contrastive decoding. We have revised the related work section to include studies relating intermediate-layer representations of vision models to EEG/MEG signals.
>
> ---
>
> ### Requested Change 2: *The authors should explain how the algorithm selects the optimal layer. It appears so far that the choice is made a posteriori*
>
> We thank the reviewer for raising this point. We sweep across layers of the vision encoder, and we have moved the related description to the Methods **Section 3.2**. The layer sweep is a systematic characterization of how the granularity of vision-model representations shifts with depth, from predominantly low-level visual features to predominantly high-level semantic features, and how this shift interacts with the information content of EEG/MEG signals. The decoding performance peaks at the layer whose granularity is best matched to the neural signal, and the table reports this peak as an indication of the method's potential, while **Fig. 2a** visualizes the full granularity–performance trajectory across layers. We agree that an a priori layer-selection criterion would be valuable, and we view it as a natural direction for future work.
>
> ---
>
> ### Requested Change 3: *It is unclear where the UMAP projection comes from. As in what is the input feature? Is it a concatenation of the decoded neural and DNN features?*
>
> The UMAP is fitted on the neural embeddings $\mathbf{v}$ and the visual embeddings $\mathbf{w}$ jointly, by stacking them as samples in the same shared embedding space rather than by concatenating them along the feature dimension. Each point in the resulting plot therefore corresponds to either a single $\mathbf{v}$ or a single $\mathbf{w}$, which allows their geometric distributions to be compared in a common 2D coordinate system. We have revised the wording to make this clearer.

---

> > ### Comment · Reviewer_9UHe · 2026-05-14
> > **Neuroscience point**
> >
> > Thank you very much for the detailed response. Before I dive into the revised paper, I have to address a point in your comment. You mention that the paper is not neuroscience focused which is understandable but I invite you to reconsider this position for the following reasons.
> >
> > The paper is dealing with brain activity data, meaning that any technological advance in analysing it has to fit within the neuroscience. You should ask yourself the questions in this sequence:
> > - Why is your work important? Because you are addressing the alignment problem between EEG/MEG data and deep learning models
> > - Why is this alignment problem important?
> > - Why is decoding vision EEG/MEG signals an important problem important to warrant you trying to improve it?
> > You only answered the first question while the other two are inherently neuroscience questions. In some cases the answer can be obvious but that's not the case here.
> >
> > Another point here is that the manuscript does not offer a new technical approach. It is a change in data target which warrants the question of why is this data a better target where your explanation of granularity is not necessarily the only explanation as I mentioned in the original comment. This warrants neuroscience understanding of what EEG/MEG signals are.
> >
> > Given your response here that EEG/MEG signals contain signals at different levels of granularity, the best approach would not be to select an intermediate layer. It would be to combine the best features across all layers using a weighted sum. Check this: https://www.nature.com/articles/sdata201912 and this: https://pubmed.ncbi.nlm.nih.gov/29756028/
> >
> > All these topics warrant a neuro understanding.

---

> > > ### Author Response · Authors · 2026-06-03
> > > **Response to Reviewer 9UHe**
> > >
> > > We thank the reviewer for the insightful comments.
> > >
> > > ### Why EEG/MEG-based decoding is scientifically meaningful
> > >
> > > Paired brain–stimulus data are inherently scarce, while image–text data are available at massive scale and have enabled powerful pretrained vision and vision–language models. Aligning neural signals with these model representations provides a way to connect limited neural recordings with large-scale visual and semantic knowledge, making decoding a bridge between neural data and pretrained large models.
> > >
> > > EEG/MEG signals offer millisecond-level temporal resolution, allowing us to examine the dynamic transition from low-level perceptual features to higher-level semantic representations during visual processing. Therefore, improving EEG/MEG-based visual decoding is not only an engineering goal, but also a way to test which computational representations best correspond to human neural responses.
> > >
> > > ---
> > >
> > > ### Additional Suggestion: *Combining the best features across all layers using a weighted sum*
> > >
> > > We appreciate the reviewer's insightful suggestion. We agree that combining information from multiple layers is a promising way to exploit complementary visual features. In the revised manuscript, we have added a new experiment on multi-layer cascade fusion to evaluate this idea (**Section 4.8**). Specifically, instead of relying on a single best intermediate layer, we select several neighboring high-performing layers around the optimal depth, independently compute their cross-modal similarity matrices, and fuse the retrieval predictions by averaging these matrices. The results show that multi-layer fusion further improves retrieval performance, with Top-1 accuracy increasing from **82.8%** using the best single layer to **90.8%** after fusing neighboring intermediate layers. This supports the reviewer's suggestion and indicates that layers provide complementary granularity for neural alignment.

---

### Review · Reviewer_X5RG · 2026-04-26

**Summary Of Contributions:**

This paper shows that neural decoding between EEG/MEG and cnn/vit embeddings works best using intermediate layers of the cnn/vit.  The paper argues that most of the EEG/MEG signal best aligns with mid-level visual artificial network embeddings, and in particular, with layers well below the last layer that is mostly semantically-aligned.  Empirical experiments show that intermediate layers do indeed work best in the task of selecting among 200 test images based on EEG/MEG recordings when subjects are presented with a test image.  This finding is consistent across a variety of different model architectures.  In addition, the paper demonstrates that larger models result in better performance on this task when using intermediate layers, even though this is not the case when using the last layer features.

The findings make a lot of intuitive sense when the cnn/vit layers are viewed as loosely mirroring the human visual system; if the EEG recordings that might correlate with a stimulus are predominantly from the visual response, as opposed to semantic recognition or other activity, one might expect best task performance using embeddings corresponding to cortex areas with high representation in the signal.  For a visual signal, these could be from anywhere that forms much of the visual processing respones in areas recorded by the EEG or MEG device, and so multiple feature layers might correspond to different parts of the signal in varying amounts.

**Additional Comments:**

* Lower layers may be affected by average pooling:  It's possible that layers even lower than those found optimal might do better than they are now with a different extraction method than average pooling.  Average pooling over a larger number of lower-level features (which may also be lower dim than intermediate) might average away relevant features for these layers.  In particular, it may be that foveal regions in the center (where the image subject is) are more represented in the neural signal, but these make up a smaller proportion of pixels compared to bg, and could be averaged away.  If this is happening, the inverted U of Fig 2a might shift to further the left if handled differently.

* Have you tried using aggregation methods other than average pooling?  A few that come to mind:  (a) pooling over each of the 9 regions of a 3x3 grid and concatenating; (b) learned spatial weighting map (i.e., learned 2d map A of size (Hl, Wl, 1) the same spatial size as features for layer l, and then average pooling features * A, or possibly using softmax(A) so param'd in logits); (c) self-weighting importance using linear proj or 2-layer MLP (so pooling features * softmax(conv1x1(features), where the conv1x1 is a 1x1 conv that outputs 1 feature dim, or using mlp instead of single linear layer).  Some of these might be a little too involved for this study, but since the paper is very much about feature level alignment, I wonder if the alignment might actually be even a little lower than what is described here, due to avg pooling artifacts.


* Is the optimal layer for the ViTs part of the transformer or earlier conv stem features?  Some of these ViTs will run a several-layer convnet first, then the transformer.  In which part is the best layer for these, and if in the transformer part, how deep into the transformer?

* Scaling laws parameter counts:  Are the params counted the total params, or only those before the intermediate layer that is used for feature comparison?  I believe it may be more appropriate to count just the params before the layer that is used, as a measure of the model capacity generating the features.  But there might also be reason to use all the params, since it measures capacity of the model whose features were trained.  It might also be informative to look at both.

* Appendix A says "For all experiments, we restrict EEG channels to those over the occipital and parietal lobes, which are closely associated with visual perception and visuospatial processing." --- What are the channels that were excluded?  And how much does restricting to these exact channels affect the findings in this paper?  Might other layers be more correlated with other channels, or other EEG that was not even part of this dataset?


* Does the best-aligned feature layer vary with subject?  Do EEG/MEG better align with some layers vs others depending on variation between subjects or recording sessions, and to what degree?  This could happen either from differences between the underlying activity or recordable signal between subjects, or from electrode placement or other session differences.


* What about using a combination of feature layers, not just one?  In particular, a selection of early, mid-level and late?  Depending on what the EEG or MEG is able to record, signals aligned with different layers might be represented.

* Table 1:  which vision model was used for this table?

* I would have liked a better background summary of the neural encoder f_theta that is used.  I'm not very familiar with current neural decoding methods, so basic summary of how the signal is preprocessed and fed to the model, along with brief architecture and size description, would be very useful in the main text.  In particular:  number of electrodes, placements relative to visual cortex areas, preprocessing transformations, time sample rates.  Most of this is in appendix A, but I think a least more information on Things-EEG and EEGProject are needed for context in Sec 3.

**Audience:**

Yes

**Audience Explanation:**

This is a good characterization of how artificial network embeddings align to EEG/MEG signals, and in particular, how intermediate layers are most represented in the recordings.

**Claims And Evidence:**

Yes

**Claims Explanation:**

Overall: Yes.  The findings are presented from multiple angles and appear quite robust overall.  The core finding that intermediate layers better align than the final layers in these data is solid.  Experiments use multiple architectures and two task setups.

A couple limitations could be better addressed:

The findings may be limited to the datasets' collection procedures.

The end result depends on the embedding extraction procedures, and in particular, the inclusion of a global average pooling step.  While this shouldn't affect the intermediate vs final layer findings that are the main thrust, it could skew the results towards higher intermediate layers rather than even lower ones than might be optimal with other embedding selections.  See additional comments for more details.

**Requested Changes:**

* Additional discussion or experiments with extraction beyond global average pooling
* Justification of why use all param counts as scaling variable, rather than just those before the feature layer --- or alternatively, using this layer.
* More detailed discussion of dataset recording setups from the prior dataset works and how these might relate to or limit the intermediate layer findings
* Clarify visual model used for table 1
* Background summary of the neural decoder model architecture in the main text would have also been useful

---

> ### Author Response · Authors · 2026-05-13
> **Response to Reviewer X5RG**
>
> We thank the reviewer for the careful reading and valuable suggestions. These suggestions helped us improve the completeness and clarity of the manuscript, and we have incorporated all corresponding revisions into the updated manuscript. We address each of the reviewer's comments in detail below.
>
> ---
>
> ### Requested Change 1: *Additional discussion or experiments with extraction beyond global average pooling*
>
> We thank the reviewer for this valuable suggestion. We have added a new analysis comparing five pooling heads (`avg`, `center`, `spatial_map`, `self1x1`, `selfmlp`), in which adaptive weighting consistently outperforms uniform averaging, with the gain most pronounced in shallow layers (e.g., **+30.6% at InternViT layer 4**). The depth of peak accuracy is not altered by the choice of pooling head, leaving the intermediate-layer conclusion unchanged. The full results and discussion are provided in **Section 4.7** of the revised manuscript.
>
> ---
>
> ### Requested Change 2: *Justification of why use all param counts as scaling variable, rather than just those before the feature layer --- or alternatively, using this layer*
>
> The parameter counts in the main text correspond to the total parameters of the vision backbone, reflecting the model capacity under which the features were trained. We agree that the parameters up to the chosen alignment layer are also informative, as they measure the effective capacity producing the features actually used for decoding. We have revised the manuscript and added an additional analysis in **Appendix B.6** that reports both counts. The scaling trend is consistent under either definition, indicating that our conclusion about model-scale–performance relationships is robust to this choice.
>
> ---
>
> ### Requested Change 3: *More detailed discussion of dataset recording setups from the prior dataset works and how these might relate to or limit the intermediate layer findings*
>
> To quantify the impact of our channel selection, we re-ran the full layer sweep with five region-based EEG subsets (frontal, central, parietal, occipital, temporal) and the full 63-channel montage, on four representative backbones (RN50, RN101, ViT-H-14, InternViT). The results (new **Appendix B.4**) showed that channels from non-visual regions (frontal, central, temporal) carry essentially no decodable signal, with top-1 accuracy peaking at less than 12% across all backbones, confirming that they do not contribute to decoding. More importantly, our intermediate-layer conclusion is unaffected by the channel selection: the inverted-U layer-wise curve peaks at the same relative depth across all configurations, including the full 63-channel montage.
>
> We also report a per-subject layer sweep on THINGS-EEG across all ten subjects and eight vision encoders (new **Appendix B.5**), and find that the best-aligned layer is largely consistent across subjects within each model: for RN50, RN101, and ViT-H-14, nearly all subjects share the same optimal layer, while larger ViT-family models show only modest variation within nearby intermediate depths. Subject variability manifests mainly as differences in overall decoding accuracy rather than shifts in the preferred representational stage, leaving our intermediate-layer conclusion unchanged.
>
> ---
>
> ### Requested Change 4: *Clarify visual model used for Table 1*
>
> The results in **Table 1** correspond to the best-performing layer of InternViT, which we now state explicitly in the second paragraph of **Section 4.3**.
>
> ---
>
> ### Requested Change 5: *Background summary of the neural decoder model architecture in the main text would have also been useful*
>
> We have added a concise description of EEGProject to **Section 3.1**. We have also expanded **Section 4.1** with the requested details on THINGS-EEG and THINGS-MEG datasets. Full specifications remain in **Appendix A**.
>
> ---
>
> ### Issue in Additional Comments 1: *Is the optimal layer for the ViTs part of the transformer or earlier conv stem features?*
>
> For every ViT backbone in our paper, all extracted "layer N" features correspond to outputs of intermediate **transformer blocks**. The optimal layer is therefore always located within the transformer (excluding both the patch-embedding stem and the final block) and consistently lies at an intermediate depth, roughly **30–65% of total transformer depth**, with deeper backbones peaking somewhat later within that range.

---

> ### Author Response · Authors · 2026-06-03
> **Response to Reviewer X5RG II**
>
> ---
>
> ### Issue in Additional Comments 2: *What about using a combination of feature layers, not just one?*
>
> In the revised manuscript, we have added a new experiment on multi-layer cascade fusion to evaluate this idea (Section 4.8). We select several neighboring high-performing layers around the optimal depth, independently compute their cross-modal similarity matrices, and fuse the retrieval predictions by averaging these matrices. The results show that multi-layer fusion further improves retrieval performance, with Top-1 accuracy increasing from 82.8% using the best single layer to 90.8% after fusing neighboring intermediate layers.
>
> ---
>
> The remaining points raised in the additional comments are addressed in our responses to Requested Changes 1–5 above.

---

> ### Comment · Reviewer_X5RG · 2026-06-07
> **responses**
>
> Thanks for your comments.
>
> The new revision addresses all of my questions and concerns well, particularly the new studies on pooling and aggregation methods.  Overall I think this is a good work with thorough measurements and profiling of feature alignment across multiple models.  The main limitation is whether it might be specific to these particular datasets and experimental protocols, but this is a minor issue as it shows clear evidence in at least two settings.
>
> Looking over the other reviews, I see 9UHe had concerns this profile might already have been performed by the community.  I was also surprised to find an apparent reliance of last-level features here, so looked into this with a quick search myself.  I did find a few additional related works, which are worth noting, but none with the same combination of focus on per-layer alignments with as robust measurements.  [1] below is closest, and has overlapping measurements; this is an apparently concurrent work on the same data and might serve as additional verification of some of the findings here.  Most or all could be included in a related work section.
>
> Note, this is just what I found quickly; based on this, I think a somewhat more comprehensive search by the authors could be appropriate, but I suspect these are some of the most related:
>
> [1]  https://arxiv.org/pdf/2603.07077v1
> [2]  https://www.nature.com/articles/s42003-026-09685-w
> [3]  https://www.biorxiv.org/content/10.1101/2025.03.11.642570v1

---

### Review · Reviewer_mE3V · 2026-05-24

**Summary Of Contributions:**

This paper is a study on the neural visual decoding from non-invasive EEG/MEG signals. Its main argument is that current contrastive decoding methods often align neural signals with the final-layer embeddings of large vision models, but those final embeddings are too semantically compressed. In contrast, neural responses retain a mixture of low-level perceptual structure and high-level semantic information.
The proposed method, Shallow Alignment, changes the alignment target from the final output of the visual encoder to selected intermediate-layer representations. The authors argue that intermediate layers provide a better granularity match because they preserve local structure while still containing useful semantic information.
Empirically, the method shows large gains on THINGS-EEG and THINGS-MEG, especially in the intra-subject setting. The paper also claims that intermediate-layer alignment restores a positive scaling relationship between vision backbone capacity and decoding performance, whereas final-layer alignment produces a “Depth-Capacity Paradox.”
Overall, the work is well motivated, easy to understand, and empirically strong. The main risks are whether layer selection is fairly validated, whether the comparison with prior methods is fully controlled, and whether the relatively modest inter-subject gains limit the broader claim.

**Audience:**

Yes

**Audience Explanation:**

The findings of intermediate layers are a better alignment target than the final layers, and are likely to be of interest to a subset of the TMLR audience, focusing on works in representation learning, contrastive alignment, transfer learning, and multimodal learning.

**Broader Impact Concerns:**

The overall concern is moderate. The main concern is that the EEG/MEG visual decoding could contribute to the privacy risks around neural data. Even though the paper focuses on image retrieval, the broader direction aims to infer brain activity contents. If this research is further improved, it might raise concerns about unauthorized decoding of brain activities, such as visual reasoning, with deep learning methods. However, the current approach does not show such capabilities and thus does not raise an immediate privacy concern.

**Claims And Evidence:**

Yes

**Claims Explanation:**

The paper framed the neural visual decoding problem as a matching problem and showed that the intermediate-layer visual representation performs better than the final-layer representations based on EEG/MEG image retrieval on the THINGS dataset.

However, for the broader claims that Shallow Alignment resolves the granularity mismatch, it is not fully proven. The experiment shows that the intermediate layers work better, but it is not a direct proof that this improvment by matching across low-level and high-level visual features. The UMAP analyses are suggestive, instead of direct evidence.

The scaling-law claim is plausible but not directly proved as well. The paper shows a positive correlation across several backbones with the optimal intermediate layers, but it is not a robust law, as the sample size of backbones is limited.

The inter-subject claim is not supported, as the THINGS-EEG dataset suggested that some subjects perform worse than the baseline. This suggests that the model is better for within-subject decoding rather than generalizing to different subjects.

**Requested Changes:**

1. Clarify the layer-selection protocol: Currently, the paper emphasizes that the intermediate layer is selected, but does not explicitly mention that the best-performing intermediate layer is selected using the validation set or test set.
2. The work emphasized that the improvement is made by the intermediate layer selection. However, the pooling analysis showed that adaptive pooling can also produce large gains. Therefore, I suggest that authors add better controlled ablations, separating layer choice and pooling choice.
3. There are other studies also working on intermediate vs. final layer alignment, and therefore, I want to suggest that the authors better position the work by comparing it with the other related works to show the originality of their contribution.
4. The current scaling laws might be too strong, as they are only based on a limited number of backbones, with different architectures and pretraining objectives. Therefore, I would like to suggest modifying the claim to avoid over-claiming.
5. Discuss the limitations of the inter-subject variability. The current work does not emphasize the inter-subject enough.

---

> ### Author Response · Authors · 2026-06-03
> **Response to Reviewer mE3V**
>
> We thank the reviewer for the careful reading and valuable suggestions. We address each of the reviewer's comments in detail below.
>
> ---
>
> ### Requested Change 1: *Clarify the layer-selection protocol*
>
> We thank the reviewer for pointing this out. We have clarified the layer-selection protocol in **Section 4.2**. Following the official dataset protocol and for fair comparison with prior baselines, the main experiments train on the provided training split and evaluate on the held-out zero-shot test split. We further added a validation-based selection analysis in **Appendix B.2**, where 74 of the 1,654 training concepts (approximately **5%**) are held out for validation. The results show only minor differences in absolute accuracy while preserving the same intermediate-layer advantage, confirming that our main conclusion is not affected by the selection protocol.
>
> ---
>
> ### Requested Change 2: *The improvement is attributed to intermediate-layer selection, but the pooling analysis showed that adaptive pooling can also produce large gains. I suggest adding better controlled ablations, separating layer choice and pooling choices*
>
> To separate the two factors, we have added **Figure 4(c)**, in which average pooling is held fixed across conditions. With pooling fixed, moving the alignment target from the final ViT output to the intermediate layer $L_{28}$ raises Top-1 accuracy from 56.7% to 82.8% (**+26.1%**); with the layer fixed at $L_{28}$, switching from average to adaptive self-$1\times1$ pooling adds only 82.8% → 87.7% (**+4.9%**). This shows that intermediate-layer selection is the dominant source of improvement, while adaptive pooling provides a complementary refinement.
>
> ---
>
> ### Requested Change 3: *Other studies also work on intermediate vs. final layer alignment; I suggest better positioning the work by comparing it with related works to show the originality of the contribution*
>
> We appreciate the reviewer's suggestion. We agree that prior studies have investigated hierarchical or intermediate visual features in neural decoding, and we have revised the Related Work to clarify our distinction. Our contribution is not only using intermediate layers, but formulating and systematically validating a **granularity-mismatch hypothesis**. Across diverse pretrained vision backbones, we show that final-layer alignment can lead to a **Depth–Capacity Paradox**, whereas granularity-matched intermediate-layer alignment recovers a positive scaling trend with model capacity. We further extend this idea through spatial pooling and multi-layer cascade fusion, showing that token-level and layer-level granularity calibration can further improve neural–vision alignment. The full discussion is provided in the extended **Section 4.8**.
>
> ---
>
> ### Requested Change 4: *The current scaling laws might be too strong, as they are based on a limited number of backbones with different architectures and pretraining objectives; I suggest modifying the claim to avoid over-claiming*
>
> To avoid over-claiming, we have revised the manuscript throughout by replacing terms such as "unlocking scaling laws" with more cautious descriptions such as "recovering a positive scaling trend."
>
> ---
>
> ### Requested Change 5: *Discuss the limitations of inter-subject variability; the current work does not emphasize inter-subject enough*
>
> We agree that inter-subject variability is an important limitation and have revised the manuscript to discuss it more explicitly. Nevertheless, the proposed strategy still shows clear generalization to inter-subject decoding. We note that strong baselines such as UBP and NeuroBridge include data augmentation or cognitive-prior mechanisms, so direct comparisons do not isolate the effect of the alignment target itself. In contrast, our controlled appendix experiments show that, when only the visual alignment target is changed, replacing the final layer with a granularity-matched intermediate layer brings robust improvements in the inter-subject setting. This suggests that Shallow Alignment remains effective across subjects, although fully addressing inter-subject variability may require additional neural-side adaptation in future work.

---

### Review · Reviewer_1TMu · 2026-06-18

**Summary Of Contributions:**

This paper studies neural visual decoding from non-invasive EEG/MEG signals, focusing on the common practice of aligning neural embeddings to the final-layer representations of pretrained vision encoders. The authors argue that final-layer visual features are overly semantic and compressed, while EEG/MEG responses preserve a mixture of low-level visual structure and higher-level semantic information. To address this mismatch, they propose Shallow Alignment, which aligns neural signals to selected intermediate-layer representations of pretrained vision models rather than final outputs. Empirically, the paper reports strong improvements on THINGS-EEG and THINGS-MEG image retrieval, and shows that retrieval accuracy tends to peak at intermediate visual layers across multiple CNN and ViT backbones. The paper further analyzes this effect through a series of supporting experiments. The main strength is the the paper being able to support it's own claim on the effectiveness of Shallow Alignment empirically. Key weaknesses include the broader “granularity mismatch” interpretation and scaling claims remain more suggestive than directly established, and the core insight that hierarchical/intermediate visual features are useful for neural decoding has substantial precedent, so the contribution should be framed as a systematic EEG/MEG retrieval study rather than as a fundamentally new alignment principle.

**Audience:**

Yes

**Audience Explanation:**

Although the core idea of intermediate layers of artificial neural networks are better in aligning to neural signals isn't groundbreaking, I believe the paper's focus on more recent architectures and how retrieval performance can scale with the capacity of foundation vision models would be of interest to TMLR's audience.

**Claims And Evidence:**

Yes

**Claims Explanation:**

The paper provides convincing evidence for its main empirical claim: intermediate-layer visual features improve EEG/MEG image retrieval over final-layer alignment across the tested datasets and backbones. Notably, the recent revisions also add useful checks on details like layer selection, pooling, subject consistency, etc. However, the evidence mainly supports the retrieval result, not the stronger claim that the gains directly prove a “granularity mismatch” mechanism. The additional claims on granularity and "granularity alignment" also isn't well defined.

**Requested Changes:**

1. The result that non-final or hierarchical DNN features can better align with neural data has substantial precedent in fMRI/MEG/EEG encoding and decoding. The manuscript should not present intermediate-layer neural alignment itself as the main novelty. Instead, it should frame the contribution as a systematic EEG/MEG retrieval characterization across modern vision backbones, with layer-wise profiling, scaling analysis, and pooling/fusion ablations. The current writing suggests that only few works have investigated aligning neural signals to intermediate layers of DNN features, which is false. (e.g. Lu et al., 2026 / ReAlnet, Communications Biology, Seeliger et al., 2018, NeuroImage, Wen et al., 2018, Cerebral Cortex)
2. The paper uses “granularity” as an intuitive representational-abstraction concept rather than as a formally defined quantity. In practice, granularity is operationalized by visual-layer depth, spatial pooling strategy, and multi-layer fusion. The alignment itself is learned through linear projections and a contrastive loss, but the key choice of the alignment layer is selected empirically through layer sweeps/validation rather than by an automatic or principled granularity estimator. This is acceptable as an empirical protocol, but the manuscript should be clearer that Shallow Alignment is partly a layer-selection procedure rather than a fully automatic granularity-matching method. I recommend the authors clearly define granularity and how it can be evaluated on DNNs or not have this term be central to the manuscript.

---

> ### Author Response · Authors · 2026-06-20
> **Response to Reviewer 1TMu**
>
> We sincerely thank the reviewer for the thoughtful and constructive comments, which helped us improve the clarity and positioning of the manuscript.
>
> ### Clarifying the Novelty Relative to Prior Neural Alignment Studies
>
> We thank the reviewer for pointing this out. We agree that non-final or hierarchical DNN features have precedent in neural encoding and decoding studies. In the revised manuscript, we have modified the related work discussion to explicitly acknowledge prior fMRI/MEG/EEG studies that mapped DNN layer hierarchies onto visual cortical processing stages. These revisions clarify that our novelty lies in systematically characterizing when and how intermediate representations improve EEG/MEG retrieval, rather than in the general observation that non-final DNN features can better align with neural data.
>
> ---
>
> ### Defining Granularity and Positioning Shallow Alignment
>
> We thank the reviewer for the constructive comment. In the revised manuscript, we define granularity as the level of visual abstraction and spatial aggregation encoded by a visual representation, covering both semantic granularity and spatial granularity. We also explicitly state that Shallow Alignment should be understood as an empirical granularity-calibration protocol based on layer selection, rather than a fully automatic granularity-matching method. We further note that developing an automatic and principled granularity-matching estimator is an important direction for future work.

---

### Author Response · Authors · 2026-06-03
**General Response**

We thank the reviewers for their helpful and constructive comments.

In this work, we study how non-invasive EEG/MEG signals can be better aligned with advanced pretrained vision models, providing a practical interface between limited neural data and the large-scale visual knowledge encoded in foundation models. We identify granularity mismatch as a key bottleneck: final-layer embeddings are often overly semantic and compressed, whereas EEG/MEG signals retain mixed structural and semantic information. Shallow Alignment addresses this mismatch by using granularity-matched intermediate representations, improving decoding performance and recovering a positive scaling trend with model capacity. In the revision, we further extend this idea through spatial pooling and multi-layer cascade fusion, showing that token-level and layer-level granularity calibration can further enhance neural–vision alignment.

The revised manuscript strengthens this framing and adds several clarifications and analyses. Specifically, we expanded the Related Work, clarified the layer-selection and UMAP procedures, softened over-strong scaling claims, and added experiments on pooling strategies, multi-layer fusion, validation-based layer selection, effective parameter counts, channel selection, and subject-level consistency.

We believe these revisions meaningfully improve the rigor and clarity of the paper, and we thank the reviewers for helping us achieve this.